# Generative emergence of non-local representations in the hippocampus

Yuchen Zhou [1], Jeremie Sibille[1] & George Dragoi [1,2,3] ✉

The role of internally-generated network dynamics in rapid temporal sequence coding, updating, and parallel recalling of alternate spatial and mental navigation contexts has remained unclear. Here, we revealed rapid emergence of temporally-compressed hippocampal theta sequences in adult male rats within 1-2 laps of a novel detour via re-purposing of pre-existing correlated neuronal sequence motifs expressed during pre-detour sleep. Detour experience-induced neuronal remapping and plasticity were consolidated and reconfigured hippocampal network during post-detour sleep, which predicted future representational drift expressed during the following post-detour reversal-track runs. Pre-detour or reversal-track representations flickered with detour representation across distinct phases of theta oscillation, revealing segregation of non-local internally-generated/recalled and local experience-related hippocampal representations by theta phase. These findings demonstrate that internal generative network dynamics across brain states support rapid theta-scale sequential coding, within-day representational updating, and flickering parallel representations − collectively forming non-local representations − during navigation.

The hippocampus is believed to support rodent navigation by creating an internalized "cognitive map" of the external environment[1,2]. An important hallmark feature of a cognitive map is in enabling animals to infer, internally generate, and perform detours and shortcuts through a familiar space beyond guidance by external landmarks and salient cues. The rodent hippocampus displays "place cells" that have selective firing at specific locations along the animal's trajectory through space[2,3]. The spatial selectivity of hippocampal activity is contributed to by multisensory external drives, which have been extensively studied by examining how place fields tune to environment geometry and distal cues[4–7]. Moreover, hippocampal place fields do not form an unbiased, uniform representation of space, since place fields tend to accumulate around goal locations[8,9], can sustain multiple maps for the same environment[10–17], and can be altered by internally driven changes in network activity[18]. This indicates the hippocampal map is not a precise reflection of the external environment but is rather contributed by internally generated representations of the external world, which depend on an animal's past and current experience, motivation,

and goals. The internal drive of hippocampal representations is further supported by the discovery of time-compressed hippocampal preplay activity patterns[19–24] and theta sequences[25–27], which support rapid and distinct encoding of multiple novel environments[20,28,29].

Temporally-compressed sequence coding in the form of hippocampal theta sequences can develop rapidly with limited experience in a novel environment[30] while the hippocampus receives multisensory external stimuli at uncompressed, behavioral timescale. Whereas whether and how hippocampal preconfiguration expressed as preplay during sleep or waking rest can facilitate rapid behavioral time scale sequential coding of novel contexts have been extensively studied using template matching[19,20,23,31–33], Bayesian decoding[20,22,24,32], and predictive coding[21,34] approaches, the direct relationship between preplay and theta sequences, both temporally-compressed phenomena, was never investigated.

The interplay between the internal and external drives on cognitive map formation and expression has received considerable attention[4,18,32,35–37]. Temporally-compressed sequential patterns of

[1]Department of Psychiatry, Yale School of Medicine, New Haven, CT, USA. [2]Department of Neuroscience, Yale School of Medicine, New Haven, CT, USA. [3]Wu Tsai Institute, Yale University, New Haven, CT, USA. ✉e-mail: george.dragoi@yale.edu

neuronal ensemble activity expressed during pre-exploration sleep form a large repertoire of pre-existing firing sequences that are selected, allocated, and partly modified during encoding of multiple possible future experiences[19,21,38]. The experience-dependent modification of internally generated patterns of hippocampal activity was described as hippocampal neural plasticity during behavior[18,39,40] and plasticity during offline replay[22,32,41–44]. Neural plasticity is essential for the operation of a cognitive map as it permits the updating of the internal model based on new experiences[21]. This plasticity also indicates that hippocampal spatial representation could change during revisits of unchanged external environments, likely due to changes in the internal state of the network acquired in between successive visits. This altered representation of the same external environment can be interpreted as within-day 'representational drift'[11,15–17,45] when its definition is broadened to allow the occurrence of different experiences between exposure and re-exposure to an unchanged familiar context. Earlier studies indicated that a novel experience could induce intra-hippocampal synaptic plasticity, mostly supported by changes in lower-activity cells or neural assemblies[18,22]. However, what remained unknown is how this plasticity will impact the future representation of a previously explored spatial context and which brain states are actively involved, given that cognitive map-related experiences, such as spatial detour and reversal run, should be able to update the internal model and thus reshape the representation of unchanged environments.

To address the question of how internal network dynamics support non-local representations, we designed a repeated detour task during exploration of a familiar maze, followed by a reversal run restoring the original maze configuration, with sleep sessions occurring before and after maze explorations. We defined non-local representations as internally generated or recalled hippocampal activity patterns that do not correspond to the animal's current location or ongoing experience, including look-ahead theta sequences, representational drift influenced by intervening experiences, and flickering between alternate contextual representations. We investigated how the hippocampal network pre-configuration supported the rapid time-compressed sequential coding of novel detour experiences, and how the novel experiences had network effects lasting into the reversal run sessions, the latter envisioned as supporting a form of representational drift. Moreover, we explored the potential parallel hippocampal co-representations of alternate non-local contexts when the internal model had richer context predictions beyond the current local external stimuli.

## Results

### Neuronal ensemble recording during a novel detour task

To study the hippocampal neuronal activity during an unexpected detour experience, we performed electrophysiological recordings from the hippocampus CA1 area in five adult rats during a task consisting of four run sessions (Run 1–4) intercalated with four sleep sessions (Sleep 1–4; Fig. 1a, b; Sleep duration: 124.0 ± 9.2 min as mean ±s.e.m.). In Run1, food-deprived rats ran on an elevated familiar square maze consisting of four connected 150 cm linear tracks (tracks 1–4, T1-T4) to collect food rewards placed on both sides of a permanent barrier separating T1 and T4. After Run1, two novel U-shape 150 cm detour segments were introduced in succession, one detour per session, on T2 and T4 during Run2 and Run3, replacing the original middle linear segment (50 cm) that was temporarily removed. The order of T2-T4 detours was counterbalanced across individual animals ("Methods"). After completion of the corresponding detour session, access to the detour segment was blocked by the original barriers, and the original middle segment was placed back. Run4 was a reversal session where all the changes were restored to the Run1 configuration. We called the 50 cm removable middle segment during the pre-detour session as the 'mobile segment' of the detoured track, and when that segment was

placed back during the post-detour session, as the 'reversal segment' relative to the detour. The start and the end 50 cm segments on the detoured tracks were not changed and were defined as 'stationary segments'. The other two linear tracks, T1 and T3, remained unchanged across the experiment. Throughout the manuscript, we used 'L' to refer to the 150 cm linear tracks without detour, and 'detour segment' to refer to the 150 cm novel U-shape detoured segment. The pre-detour session referred to the run session immediately preceding the detour session for a given track, and the post-detour session referred to the run session immediately following the detour session for a given track.

Sleep sessions (Fig. 1a) occurred before all explorations (Sleep1), between Run1 and the detour sessions Run2-3 (Sleep2), between the detour sessions and the reversal Run4 session (Sleep3), and after all Run sessions (Sleep4). During waking rest (animal velocity < 2 cm/s) and sleep states, we detected epochs with strong multi-unit activities preceded and followed by reduced neuronal activity called frames[20,32], where we investigated preplay and replay activity. Rat 4 was excluded from all sleep analyses because of limited detected frames due to low neuronal counts and synchrony during sleep (frame numbers across Sleep1–4 combined: Rat1: 14,816; Rat2: 9718; Rat3: 7423; Rat4: 376; Rat5: 7643).

### Strong remapping between pre-detour and detour sessions

Place maps were computed based on spiking activities of putative pyramidal cells during active maze exploration (animal velocity > 10 cm/s). Pyramidal cells exhibited similar levels of spatial tuning between the linear tracks and detour segments (Fig. 1c left; T1T3L vs. DetSeg, $P = 0.8683$; T2T4L vs. DetSeg, $P = 0.1333$; Kruskal-Wallis with Tukey-Kramer; $n = 1354, 1037, 464$ for T1T3L, T2T4L, and DetSeg). However, place cells had significantly higher firing rates and place maps were more similar between the two directions on the detour segments than on linear tracks (Fig. 1c middle and right; $P < 10^{-8}$ between all pairs; Kruskal-Wallis with Tukey-Kramer; $n = 850, 643, 259$ for T1T3L, T2T4L, and DetSeg during computing bi-directionality). By introducing the unexpected novel detour experience, the animals explored two spatially different but related maze segments in different sessions: the 150 cm detour segment and the 50 cm mobile middle segment of the detoured track. We investigated the detour-induced process of place cell remapping at the single-cell, neural population level, and the sequential coding level.

At the single cell level, we tested three hypotheses for possible structured remapping (see "Methods") during detour: dominated by prospective/retrospective coding[12] based on the distance to the nearest maze corner (topological; Fig. 1d, 2nd and 4th columns), dominated by track orientation[34] (geometric; Fig. 1d, 3rd column), or stretching the spatial tuning from the mobile segment about three times to map the entire detour segment[28] (sequential; Fig. 1d, 5th column). We computed the individual neuron-level place map similarity between corresponding maze segments across pre-detour and detour sessions, controlled by changes in mapping across these sessions on unchanged maze segments. The correlation values were compared to cell-ID shuffle datasets, where across-session cell identities were mismatched. We found that the place map similarity between the pre-detour mobile segment and the detour segments were not significantly different from the shuffle cases. Meanwhile, on the stationary first and last 50 cm segments of the detour tracks (Fig. 1d, 1st and 6th columns) and the middle 50 cm segments or the entire 150 cm linear segments of non-detoured tracks 1 and 3 (Fig. 1d, 8th and 9th columns), the across-sessions place map similarities were significantly higher than the cell-ID shuffle (Fig. 1d; Direction 1, Data vs. shuffle, P-values for columns 1-9: $2.2 \times 10^{-6}$, 0.5330, 0.6720, 0.8413, 0.2710, $1.5 \times 10^{-5}$, 0.5272, $5.7 \times 10^{-12}$, $1.5 \times 10^{-23}$; $n$ values for columns 1–9: 167, 195, 195, 201, 236, 158, 271, 163, 146; Direction 2, Data vs. shuffle, P-values for columns 1-9: $2.9 \times 10^{-8}$, 0.5055, 0.1072, 0.5005,

0.5716, 6.2x10$^{-4}$, 0.2540, 4.8x10$^{-24}$, 7.5x10$^{-17}$; $n$-values for columns 1–9: 179, 207, 198, 198, 240, 169, 274, 171, 151; Wilcoxon rank-sum tests). These results indicate the detour-induced global remapping did not simply follow our hypothesized topological, geometric, or sequential structured remapping scenarios.

Next, we asked whether some of the place maps and neurons were specifically repurposed from the removed segment to the detour segment. Thus, we investigated whether the active place cells on the pre-detour mobile segment were more likely to be recruited in representing the detour segment. We found that neuronal activity on the pre-detour mobile segment was positively correlated with activity on the detour segment; however, this network preference was not significantly stronger for detour compared to other middle-track segments (Fig. 1e and Supplementary Fig. 1; $P$-values of Mobile vs. Opposite track, T1, and T3: 0.3039, 0.0636, 0.3642; Kruskal-Wallis with Tukey-Kramer). Therefore, the positive correlation primarily reflected the difference in the overall activity level between cells across track-segments rather than a preferential recruitment of pre-detour place cells of the mobile segments on the detour segments.

At the neuronal ensemble level, a population vector (PV) was constructed as all cells' firing rates at one specific spatial bin on the maze, and the cosine similarity of population vectors were computed

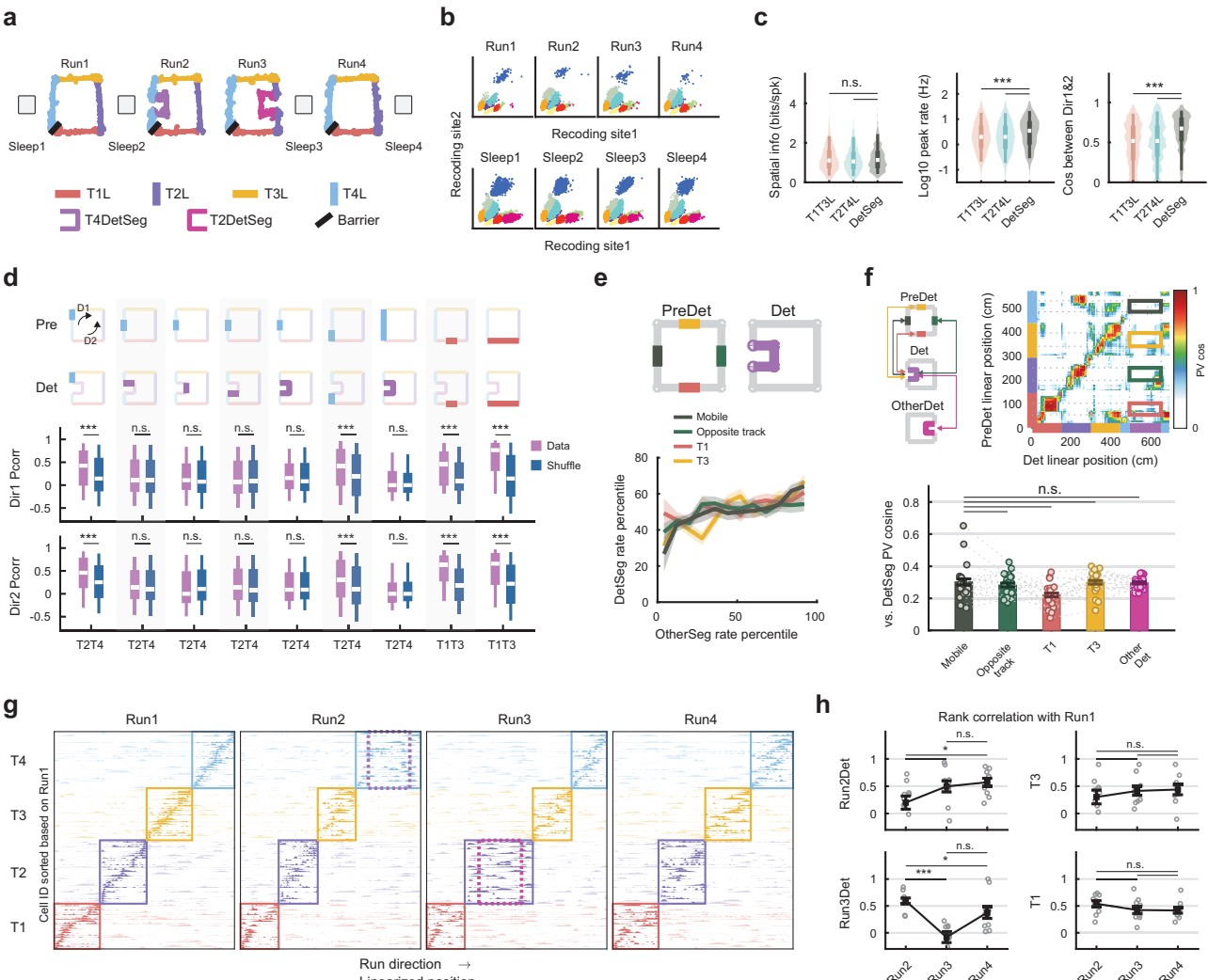

**Fig. 1 | Detour task and hippocampal remapping during detour. a** Diagram of experimental setup with 4 run sessions (Run 1–4) alternating with 4 sleep sessions (Sleep 1-4). T1-T4L: linear tracks when not detoured. **b** Example stable spike clusters from one tetrode across all sessions. **c** Place cells spatial tuning properties on detour and linear track segments. T1-T4L: linear tracks when not detoured. Kruskal-Wallis with Tukey-Kramer. ($n = 1354, 1037, 464$ for T1T3L, T2T4L, and DetSeg in firing rates and spatial information; $n = 850, 643, 259$ for T1T3L, T2T4L, and DetSeg in bi-directionality). **d** Place map similarity between various segment pairs from pre-detour and detour sessions. Cartoon: 9 configurations (columns) comparing segments bolded in pre-detour (top row) and detour (2nd row) sessions. Arrows mark the 2 running directions (D1/Dir1, direction 1 and D2/Dir2, direction 2). Place map similarities (Data) were compared with corresponding cell-ID shuffles (Shuffle; 3rd and 4th rows). Wilcoxon rank-sum tests. ($n = 167, 195, 195, 201, 236, 158, 271, 163, 146$ for direction 1; $n = 179, 207, 198, 198, 240, 169, 274, 171, 151$ for direction 2). **e** Firing rate similarity between detour segments and middle segments from the pre-detour session. We used the firing rate percentile among all the cells rather than the absolute firing rates to better visualize the distribution. Lines and shaded areas indicate mean and s.e.m. **f** Example population vector (PV) cosine similarity across spatial bins between pre-detour and detour sessions (top). PV similarity against detour segments from different segments (bottom) with each dot representing one animal, direction, and detour session ($n = 20$). Kruskal-Wallis with Tukey-Kramer. **g** Example place map sequence across individual T1-4 (continuous line rectangles) sorted based on Run1. Dashed line rectangle: detour segments. **h** Spearman rank-order correlation of place map sequences of Runs2–4 against Run1 for individual tracks, with each dot representing one animal and direction ($n = 10$). Student's $t$ test. Error bar plots were represented as mean ± s.e.m. In box plots, whiskers represented the 5th to 95th percentile range; the box represented the 25th to 75th percentiles, and the center dot indicated the median value. ***$P < 0.001$, *$P < 0.05$, n.s. = not significant. Source data are provided as a Source Data file.

between spatial bins across different sessions. The PV similarity between spatial bins on detour segment and pre-detour mobile segment was not significantly higher than that between detour and middle segments of tracks 1 and 3 (T1, T3; not detoured), the opposite track (OT; not detoured), or another detour segment (Fig. 1f; P-values of Mobile vs. OT, T1, T3, and AnotherDet: 1, 0.1030, 0.8517, 0.9546; Kruskal-Wallis with Tukey-Kramer).

To study the behavioral scale sequential coding, we sorted the cells based on their peak firing rate on each track during Run1, and plotted the place maps across the run sessions. The place cell sequences experienced global reorganization on the detour tracks during detour sessions (Fig. 1g). The sequence reorganization during detour was quantified by a rank order correlation with session 1, where the detour session had the lowest correlation (Fig. 1h; Two-way ANOVA, detour impact, $F(1115) = 30.0623$, $P = 2.5 \times 10^{-7}$).

### Rapid emergence of novel detour theta sequence was compatible with detour preplay

Since the experience of detour induced global remapping, we were interested in studying how rapidly the hippocampus can form a sequential representation of a novel spatial context. We observed a rapid emergence of theta sequences on the detour segments, as early as within the first detour lap, as measured by the quadrant ratio from decoding and the cross-correlogram (CCG) bias across the time-compressed and behavioral timescales[25] (Fig. 2a–d; P-values of quadrant ratio from lap 1 to 3: $8.6 \times 10^{-9}$, $1.9 \times 10^{-25}$, $6.0 \times 10^{-31}$; One side Wilcoxon signed rank test with positive quadrant ratio; P-values of CCG temporal bias correlation across time scales from lap 1 to 3: $1.4 \times 10^{-9}$, $1.1 \times 10^{-15}$, $1.8 \times 10^{-33}$; P-values of correlation between CCG temporal bias and place map bias[25] from lap 1 to 3: 0.8020, 0.0092, $9.0 \times 10^{-6}$). The expression of the theta sequence was faster than previously reported[30,46], likely contributed by the strong bidirectionality of spatial tuning on the detour segment. Since no external stimuli were presented to the animals at the theta time scale, we asked what could drive the rapid expression of theta sequences at an order of magnitude compressed time scale compared with place cell sequences. Hippocampal pyramidal cells are known to exhibit single-cell temporal coding[47] in the form of theta phase preference and phase precession of spikes[48], which could contribute to the expression of theta sequences, an ensemble temporal code. However, we found the theta phase sequence formed by place cells with significant theta phase locking did not match their novel detour place map sequence (Supplementary Fig. 2e; 1 out of 20 samples was significant, $P = 0.6415$; Binomial test with 5% chance level), and the decoded probability within a theta cycle based on spike phase histograms did not depict theta sequence structure (Supplementary Fig. 2e; $P = 0.1125$; One side Wilcoxon signed rank test with positive quadrant ratio). Similarly, while theta phase precession could contribute to theta sequence, we found that time-jittering of place cell spike times, revealed a regimen ($\pm 35$ ms to $\pm 55$ ms) where theta phase precession were abolished while theta sequences in the first two detour laps were preserved (Supplementary Fig. 2f, g; P-values of theta sequence at jittering scale of $\pm 35$, $\pm 40$, $\pm 45$, $\pm 50$, and $\pm 55$ ms: $2.1 \times 10^{-11}$, $2.0 \times 10^{-7}$, $3.3 \times 10^{-4}$, $3.5 \times 10^{-5}$, $2.9 \times 10^{-6}$; P-values of theta phase precession at jittering scale of $\pm 35$, $\pm 40$, $\pm 45$, $\pm 50$, and $\pm 55$ ms: 0.0755, 0.0755, 0.2642, 1, 1; Binomial test with 5% chance level). Both theta sequence quadrant ratio and phase precession slope were individually correlated with the jittering scale (Quadrant ratio vs. jittering scale: $R = -0.57$, $P = 2.6 \times 10^{-27}$; Phase precession slope vs. jittering scale: $R = 0.33$, $P = 3.4 \times 10^{-9}$; Pearson correlation), but they were not directly correlated, as measured using partial correlation analysis and considering jittering as a confounding factor (Supplementary Fig. 3; P-value of partial correlation: 0.274). In addition, when we eliminated single cell theta phase precession by removing spikes outside the largest burst/lap/place field (i.e., the remaining spikes had max interspike interval $\leq 20$ ms), we could still

observe theta sequences in the first 2 detour laps (Supplementary Fig. 4; After spike removal, Lap1, $P = 1.4 \times 10^{-4}$; Lap2, $P = 0.0012$; One side Wilcoxon signed rank test with positive quadrant ratio). This highlighted the critical contribution from precise temporal coordination at the neuronal ensemble level toward theta sequence expression. Our results are consistent with previous studies that found a dissociation between theta phase precession and theta sequence[25,30,49], and suggest that although theta rhythm-related single-cell properties could contribute to the expression of theta sequence, the latter cannot be fully explained by the individual cells' theta phase locking or precession.

An alternative scenario would posit that hippocampal waking rest replay during the early laps of detour could have actively compressed and consolidated the behavioral timescale experience into theta timescale sequences[50,51]. For that to be the case, the theta sequence would need to emerge only after the hippocampal waking rest replay of detour. However, within our detected hippocampal spiking activities, the emergence of theta sequence during detour run was significantly earlier than the sequential trajectory representation during waking rest replay when decoded either using clustered or cluster-less[52] spiking activities (Supplementary Fig. 2a–d; Clustered spikes decoding: Time, $P = 9.7 \times 10^{-4}$; Laps, $P = 0.002$; Cluster-less decoding: Time, $P = 4.4 \times 10^{-4}$; Laps, $P = 1.2 \times 10^{-4}$; Wilcoxon signed rank test). In the detection of theta sequences and replay, we matched the detection false positive ratios and the number of detections, and our result was robust across different detection criteria ("Methods", Supplementary Figs. 5 and 6). While the occurrence of waking rest replay depends on task design and the number of rests the animals take, here, replay statistically expressed later than the theta sequence during animals' spontaneous detour behavior, indicating it was not causal nor necessary for theta sequence time-compression. However, expression of time-compressed detour representation could happen and was observed during waking rest in pre-detour run sessions or before the first lap in the detour session (i.e., preplay) in rare cases (Supplementary Fig. 7).

Therefore, we hypothesized that network pre-configuration into sequential motifs, that are predictive over selection and allocation of future place cell sequences on novel tracks[20,21,34], could also contribute to the rapid theta-scale sequential coding during the novel detour experience (Fig. 2e). The internal pre-configured structure of CA1 network has been revealed by the discovery and the study of hippocampal preplay[19–24,32,38]. We found a significant preplay of the detour experience, where the decoded trajectory could cross several 90° detour corners. The detour preplay was significant based on absolute weighted correlation measure as well as based on combination criteria of absolute weighted correlation and maximum jump distance[32,51] (Fig. 2f, g; $P = 4.4 \times 10^{-4}$; Wilcoxon signed rank test). The expression of detour preplay occurred during both sleep and waking rest brain states, with a relatively stable preplay distribution over time spanning over 5-6 h before the detour experience, without obvious recency effects (Fig. 2h; Preplay proportion vs. Time to detour: $R = -0.0082$, $P = 0.8989$; Pearson correlation).

Next, we wanted to investigate whether the early emergence of the theta sequence could be derived from the hippocampal pre-configuration. Given that hippocampal preplay is a primary expression rather than the underlying mechanism of pre-configuration, we used a computational and analytical approach to test whether early theta sequence is compatible with pre-configuration. Previous studies used a single maze unidirectional running task and found that, using spike cross-correlogram (CCG) analysis that averaged neuronal activity temporal bias across entire run or sleep sessions, pairwise temporal structure during running theta state correlated with post-run, but not with pre-run sleep[53,54]. Using combined multiple novel detour tracks and both running directions, each with distinct CCG patterns (Supplementary Fig. 8a–c), here we showed that temporal bias computed

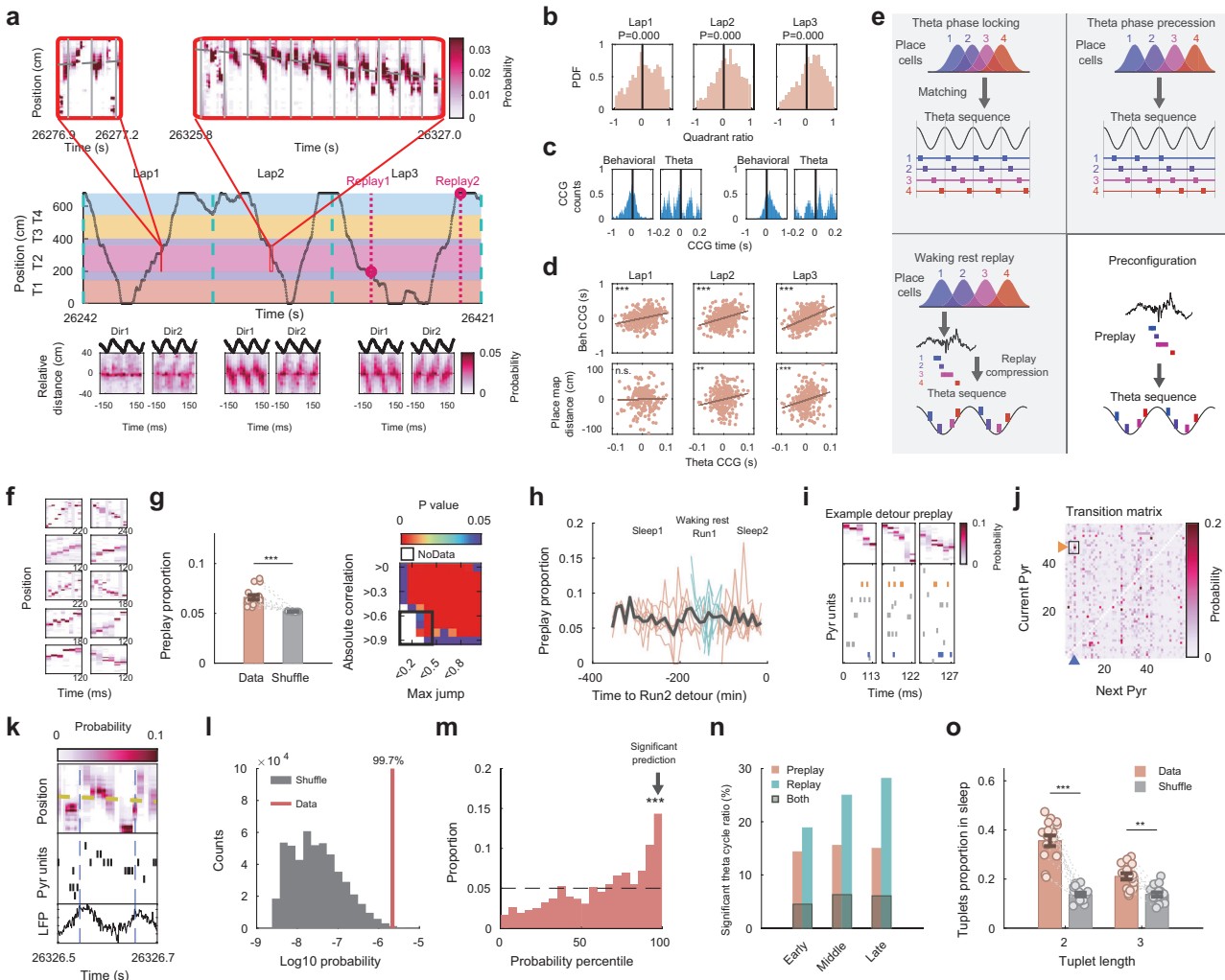

**Fig. 2 | Rapid expression of detour theta sequences is compatible with network pre-configuration. a** Example showing rapid expression of detour sequence. (Top and middle) The animal's position during the first 3 laps with zoomed-in windows showing theta-scale decoding results. Vertical lines: theta peaks. (Bottom) Decoding results averaged over detour theta cycles from 2 run directions on the first 3 laps. **b** Distribution of detour segment theta cycle decoding quadrant ratio across first 3 laps (Wilcoxon signed rank test). **c** Behavioral and theta time-scale cross-correlograms (CCG) from two example cell-pairs during the lap 1 detour run. **d** Correlation of CCG bias between theta time-scale and behavioral time-scale (top) or bias of place map (i.e., distance between place fields) on detour (bottom) across first 3 laps. Dots: cell pairs. **e** Four hypothesized mechanisms contributing to the rapid development of theta sequence: Spike theta-phase locking (top left); Spike theta-phase precession (top right); Waking-rest replay compression (bottom left); Pre-configuration (bottom right). The first 3 mechanisms were either not occurring or could not fully explain the early-laps theta sequence (Supplementary Fig. 2); thus, we focused on pre-configuration. **f** Examples of detour preplay during pre-detour sleep. **g** Proportions of significant detour preplay measured by weighted correlation (left; Wilcoxon signed rank test) or a combined criteria of absolute weighted correlation and normalized maximum jump (right; Z-tests for two proportions) compared against time bin shuffles. On combined criteria, the black box indicates the region of significant preplay. **h** Proportion of significant detour preplay in sleep and run sessions before Run2 measured in 10 min time windows.

Orange and blue lines: sleep (orange) and waking rest (blue) preplay proportions for each animal and direction. Black line and shaded area: mean and s.e.m. **i** Example frames detected as forward detour preplay of a detour segment with highlighted activities from two example cells. **j** Transition probability matrix estimated from significant forward detour preplay. Pixels indicate the conditional probability of one pyramidal cell firing (x-axis) right after another one (y-axis); one example pixel indicated by the small square and arrowheads, color-coded as corresponding cells in (**i**). **k** Example theta cycle depicting a detour theta sequence. **l** Probability of spikes in theta cycle from (**k**) computed based on detour preplay transition probability matrix from (**j**) compared against 500,000 shuffles with the same length to calculate percentile value. **m** Distribution of detour theta cycle (> 3 active cells) probability percentile values against shuffle. Significant (> 95% of shuffles) proportion of theta cycles was above chance level (Binomial test). **n** Forward preplay had steady prediction power in predicting theta cycles from the first, middle, and the last 2 laps, while prediction power from forward replay accumulated over experience. **o** Recruitment of tuplets with length of 2 or 3 cells from pre-detour sleep into early theta cycles was significantly higher than from shuffled sleep (Wilcoxon signed rank test). Bar plots in (**g, o**): mean ± s.e.m with each dot representing one animal, direction, and detour session (n = 16). ***P < 0.01, **P < 0.01, *P < 0.05, n.s. = not significant. Source data are provided as a Source Data file.

from the entire sleep CCG cannot correlate with any single experience, even during post-run sleep (Supplementary Fig. 8g). However, when we selected sleep frames that were forward p/replays for a given detour and direction experience, the same cell-pair could exhibit distinct CCG patterns within different subgroups of frames p/replaying the other detour and/or run direction (Supplementary Fig. 8d–f); their

temporal bias was indeed correlated with the corresponding run bias (Supplementary Fig. 8h). This indicates that in previous studies, the single novel run experience dominated the post-run sleep session, while single-experience-induced increase in firing rates and strengthening of cell-assembly organization likely enabled the detection of stronger temporal bias during post-run compared to the brief

(15 min-long), sparser pre-run sleep[32,55,56]. Therefore, when investigating exposure to several novel experiences (e.g., multiple tracks and/or running directions), where different experiences induce specific changes in the network, we need to either consider the inherently rich and complex sleep dynamics or adopt analysis methods with individual sleep frame resolution.

We used a Markov model built from the spiking activity during the significant detour preplay frames during pre-detour sleep to predict the spike activities during theta cycles within the first 2 detour laps. Based on spike activities expressed during forward preplay of detour during sleep (Fig. 2i), we constructed a transition probability matrix, which gave the probability of the next active cell conditioned by the activity of the current cell[21] (Fig. 2j). Using the transition probability matrix, we computed the probability of spike sequences occurring in the early laps' theta cycles with more than 3 active cells (Fig. 2k). We defined that theta sequences on the detour segment can be significantly predicted by the detour sleep preplay if the computed sequence probability was higher than the 95th percentile of the 500,000 shuffles where random sequences with the same cell numbers were generated (Fig. 2l). We found that spiking activity in the early-laps' theta cycles on the novel detour could be significantly predicted by the detour preplay activity (Fig. 2m; 14.34% theta cycles were significant with 5% chance level, $P = 4.2 \times 10^{-15}$; Binomial test). The prediction power was specific to this detour experience and forward theta sequence. Indeed, the prediction was not significant if we only used theta cycles with low quadrant ratios, theta cycles with high quadrant ratios for non-detoured tracks 1 and 3, pre-detour sleep frames that were not detour preplay, or if we conducted a cell ID shuffle in the detour preplay transition matrix (Supplementary Fig. 9; Respective $P$-values: 0.2729, 0.1386, 0.1246, 0.6227; Binomial test with 5% chance level). A compelling argument for the proposed role of preplay in driving theta sequence is that phase precession (a single neuron feature) plays no role in selecting which group of several distinct neurons will be activated within a theta cycle together as a theta sequence (a feature of the neuronal ensemble). Instead, preplay and tuplet analyses indicated this neuronal selection/grouping was already present within and could be predicted from selected frames of corresponding preplay activity.

Interestingly, forward detour preplay had a steady prediction power over subsequent theta cycle activities, from early to late detour laps. However, if we built the Markov model from forward detour replay occurring after the detour experience, the model's prediction power increased as a function of experience, and always exceeded the prediction power from preplay (Fig. 2n; Preplay prediction early laps vs. late laps $P = 0.7783$; Replay prediction early laps vs. late laps $P = 6.0 \times 10^{-4}$; Z-tests for two proportions). This result implies the pre-configuration had a steady contribution as a backbone, while the plastic impact from experience accumulated over time. This analysis is distinct from but consistent with previous research showing that preconfiguration supported behavioral timescale future spatial sequences using the Bayesian decoding method. Here, we investigated the compatibility between compressed temporal sequence during pre-run sleep and run theta cycles, based on neuronal sequence activity using mean neuronal spike times. Importantly, our analysis did not prove an absolute prediction of a novel theta sequence, since we used the future information of the detour place map to select the corresponding sleep frames. This selection procedure was necessary considering the large repertoire of sequential activity patterns expressed during sleep[29] and the variability of theta sequences during run[34] (Supplementary Fig. 8a–f). Thus, the "prediction power" in our analysis revealed the compatibility between the corresponding preconfigured frames of activity during sleep and the future time-compressed theta sequence, rather than an absolute prediction of the theta sequence from indiscriminate sleep patterns. To avoid a potential technical circularity, we selected the corresponding preplay frames using Bayesian

decoding (i.e., using all neuronal spikes within frames and neuronal coactivation) but used a Markov model (i.e., using the center-of-mass spike time/neuron and neuronal order) to compute theta sequence prediction. These two methods are based on different aspects of spiking activity patterns during sleep as well as run, as emphasized by the dissociation between detection of significant p/replay frames using decoding vs. rank order correlation methods (Supplementary Fig. 10f, g).

The Markov model was dependent both on neuronal firing rates and spiking order during sleep, thus it predicted both the allocation of cells and their relative order during run theta cycles (Supplementary Fig. 10a–c). To evaluate the spike order compatibility between pre-run sleep and early detour lap theta cycles, we conducted three analyses, which were independent from firing rates. First, we computed the pyramidal cell pair-wise spike order probability matrix based on forward detour preplays, defined as the probability that one cell fires before another given both are active in a sleep frame. Then, based on this sleep probability matrix, we estimated the spike order probabilities during early detour lap theta cycles by averaging the order probabilities for all cell pairs, and found the average probabilities were significantly higher than a 50% chance level (Supplementary Fig. 10d; $P = 0.0262$; Wilcoxon signed rank test). Second, we computed rank order correlation between spiking in each pre-detour sleep frame and early detour lap theta cycles when there were at least 5 common active cells (to have 5! = 120 independent permutations). We defined a pair to be significantly correlated if the correlation value was larger than 95% of the shuffle, and we found that among all the pairs with at least 5 common active cells, the ratio of significant correlated pairs was significantly higher than 5% chance level (Supplementary Fig. 10e; $P = 0.0097$; Wilcoxon signed rank test against 5%). Similar with Bayesian decoding, this rank-order correlation analysis had single-frame resolution to investigate how preconfiguration could support various future experiences. Last, we conducted a tuplet analysis exploring the biological mechanism of how pre-configuration could support the rapid expression of the theta sequence. Previous studies found that pre-configured high-repeat short neuronal tuplet (sequences of $3 \pm 1$ cells) motifs expressed during pre-novel run sleep were significantly recruited and allocated to the run place cell sequence[21,34]. Similarly, we found that pre-detour sleep tuplets that contributed to the detour preplay were preserved and significantly recruited into the early laps' theta sequence (Fig. 2o; Tuplets with length of 2 cells $P = 4.4 \times 10^{-4}$; tuplets with length of 3 cells $P = 0.0038$; Wilcoxon signed rank test against shuffle sleep). The tuplet analysis is different from pairwise CCG analysis since the former considers not only the neuronal temporal order but also the tendency of neuronal pairs to be active within the same frame (Methods) and potential future runs. In tuplets with a length of 2 cells (e.g., neurons A and B), around 36% of them were bidirectional (i.e., both A → B and B → A were detected as tuplets), which significantly enriched the bidirectional run sequential structures during detour run (Supplementary Fig. 11), impacting the power of CCG analysis in determining neuronal temporal bias. Our results suggest that pre-existing short sequence motifs contributed a backbone for the rapid expression of time-compressed sequential coding during novel detour exploration in conjunction with plasticity-driving inputs from presumed specific sensory-motor external stimuli.

### Impact of detour expressed during offline rest and sleep states

Since the novel detour induced strong place cell remapping, we next investigated whether this detour experience could alter the hippocampal network more persistently. We investigated hippocampal inferred plasticity (synaptic and/or intrinsic) expressed at the circuit level as changed spatial representation and reorganization of cell assemblies across brain states. We found the novel detour experience led to significant plasticity expressed during subsequent waking rest and sleep states. In both waking rest and sleep states, p/replay

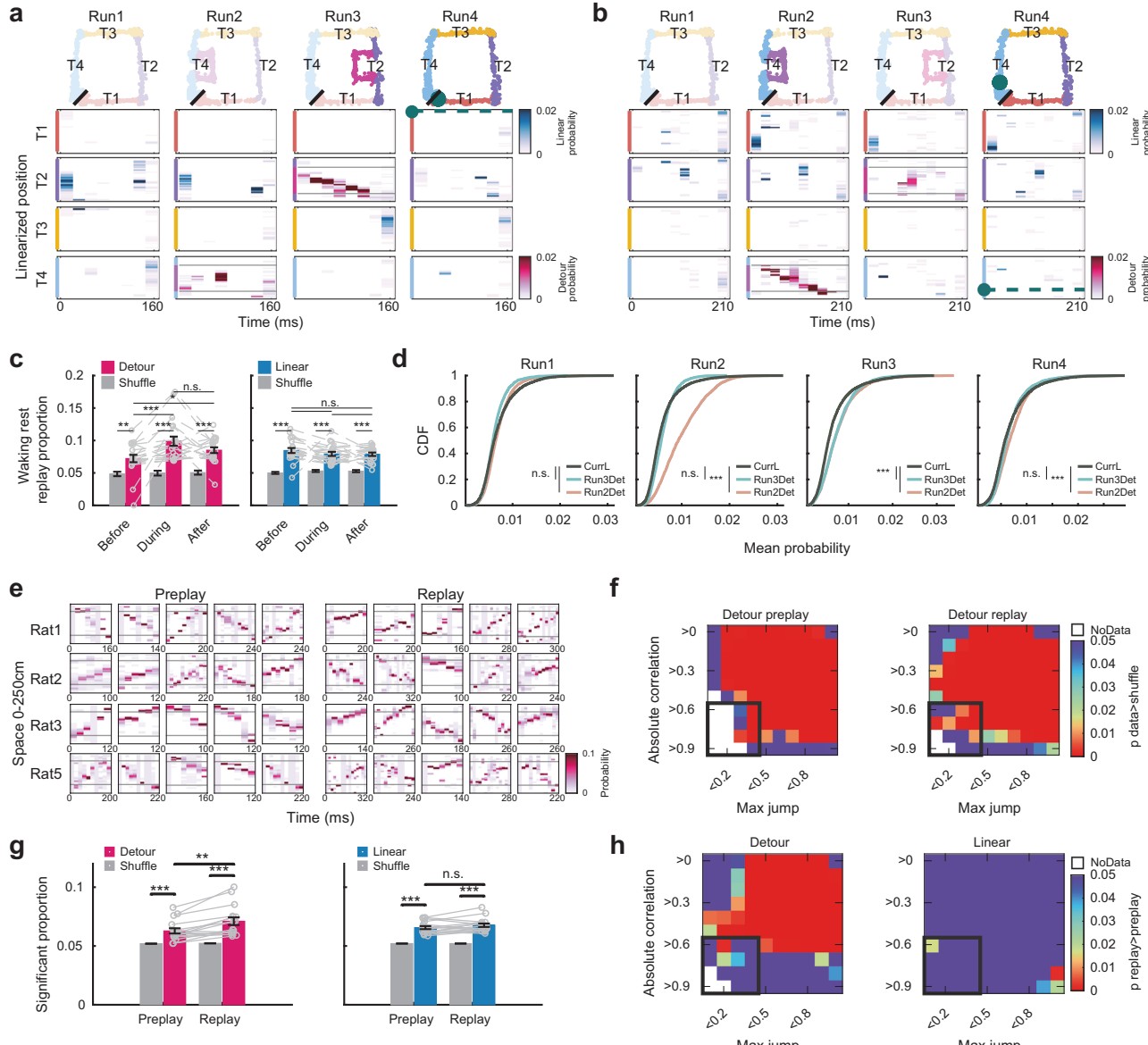

**Fig. 3 | Detour-induced neural plasticity expressed during offline states.**
**a**, **b** Example significant detour waking rest replays of T2 (a) and T4 (b) occurring after the detour session ended. Top row, cartoon highlighting the current run session and the track being replayed. Green dots mark the animal's actual position during replay. Bottom 4 rows display decoding results using place maps concatenated across tracks and sessions. Colormaps are in red for detour tracks and blue for linear tracks. **c** Proportions of significant waking rest replay of detour (left) and linear (right) tracks grouped by before-experience (frame session earlier than place map session), during-experience, and after-experience sessions. Wilcoxon signed rank test. **d** Decoding probability of the 2 detour segments and current linear tracks across sessions (Run2Det, Run3Det: detour segments in Run2, Run3;

CurrL: current session linear tracks). Wilcoxon ranksum test. **e** Examples of significant sleep preplay and replay of detour. **f** Detour preplay and replay measured by absolute weighted correlation and normalized maximum jump, compared with time bin shuffle. Z-tests for two proportions. **g** Ratio of significant preplay or replay of detour and linear tracks measured by weighted correlation compared with time bin shuffle. Wilcoxon signed rank test. **h** Replay over preplay plasticity measured with absolute weighted correlation and normalized maximum jump for detour and linear tracks. Z-tests for two proportions. Bar plots display mean±s.e.m., with each dot representing one animal, direction, and detour session ($n = 16$). ***$P < 0.001$, **$P < 0.01$, *$P < 0.05$, n.s. = not significant. Source data are provided as a Source Data file.

generally tended to be confined to individual tracks and avoid crossing maze corners (Supplementary Fig. 12), suggesting a maze representation segmentation by its corners.

During the waking rest state, we observed significant detour replay after detour sessions ended (Fig. 3a, b), even when the replayed detour segment was not currently accessible. This observation matched previously reported remote waking rest replay[57] and indicated the strong impact of a novel detour on the hippocampal network. During waking rest state, we observed significant compressed representations of detour and linear track experiences before, during, and after the actual depicted experiences occurred (Fig. 3c; Detour before,

during, after, $P$-values: 0.0036, $8.9\times10^{-5}$; $8.9\times10^{-5}$; Linear before, during, after, $P$- values: $8.9\times10^{-5}$, $8.9\times10^{-5}$, $1.2\times10^{-5}$; Wilcoxon signed rank test). Detour experience also induced plasticity on the detoured tracks (Fig. 3c; Detour before vs. during, $P = 4.2\times10^{-4}$; Detour before vs. after, $P = 0.0351$; Detour during vs. after, $P = 0.0808$; One-sided Wilcoxon signed rank test with hypothesized order before < after < during) but not on the non-detoured linear tracks (Fig. 3c; Linear before vs. during, $P = 0.9825$; Linear before vs. after, $P = 0.9347$; Linear during vs. after, $P = 0.5223$; One-sided Wilcoxon signed rank test with hypothesized order before < after < during). The decoding probability during waking rest frames was also biased towards depicting the

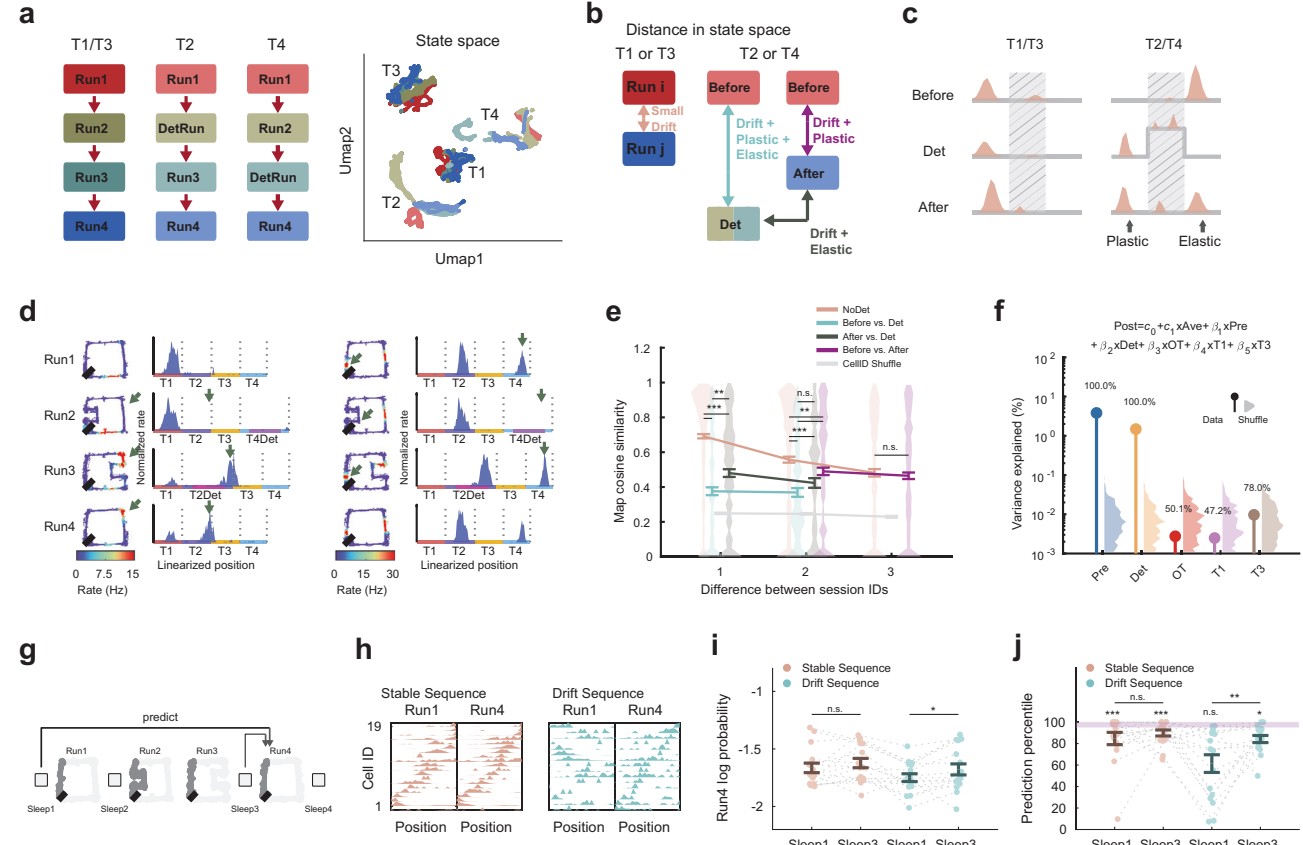

**Fig. 4 | Detour induced plasticity expressed during post-detour run was predictable from interleaving sleep. a** UMAP visualization of hippocampal activities on non-detoured (T1 and T3) and detoured (T2 and T4) tracks across sessions. **b** Detour model illustrating plastic and elastic impact of detour on future runs. Detour experience caused global remapping from before the detour session to the detour session that was partially retained (plastic) and partially restored (elastic) in the after-detour run session. **c** Cartoon illustrating the plastic and elastic effects of detour experience at the single cell level vs. control not-detoured tracks. Spatial tunings were compared across sessions at two stationary segments to compute the cosine similarity of place maps. Middle segments were excluded to ensure the same vector length. **d** Example of plastic (left) and elastic (right) detour impact on place maps. In the post-detour session, plastic place fields retained the firing pattern gained during detour; elastic place fields restored their pre-detour firing pattern (green arrows). **e** Cell's place map similarity across session pairs classified by

sessions' difference and the relationship to detour. Wilcoxon rank-sum test. (Degree of freedom 2641). **f** The variance of post-detour tuning curves on stationary segments explained by pre-detour and detour sessions' tuning curves. Listed values represented percentiles compared against shuffle. **g** Using Sleep 1 or Sleep 3 to predict place map sequence during reversal Run4. **h** Example of stable (left) and drift (right) place map sequences in Run1 and Run4. **i** Normalized probabilities of Run4 stable and drift place map sequences predicted from Sleep1 and Sleep3. Wilcoxon signed rank test. **j** Probability percentiles of Run4 stable and drift place map sequences compared against shuffle sequences predicted by Sleep1 and Sleep3. Purple region indicates significant (> 95% of shuffle) sequences. Wilcoxon signed rank test. Data are displayed as mean±s.e.m. In (**i**) and (**j**), each dot represents one animal, direction, and detour session (*n* = 16). ***P < 0.001, **P < 0.01, *P < 0.05, n.s.=not significant. Source data are provided as a Source Data file.

detour segments during and after, but not before, the detour experience (Fig. 3d; two-way ANOVA, detour segment probability * detour experience, F(2,75442) = 1285.3, P = 0).

We envisioned network plasticity during sleep as the improvement in detour representation from preplay to replay[22,31,32,41,44] (Fig. 3e). There were more significant detour preplay and replay events compared to time-bin shuffles as shown with either 2-parameter (weighted correlation and normalized maximum jump distance) (Fig. 3f) or one-parameter (weighted correlation) significance measures (Fig. 3g; Detour data vs. shuffle in preplay P = 4.4x10⁻⁴; Detour data vs. shuffle in replay P = 4.4x10⁻⁴; Wilcoxon signed rank test). With the one-parameter measure, detour representations were significantly stronger in post-detour sleep sessions compared to pre-detour sleep sessions, while plasticity was not significant for representations of non-detoured, unchanged linear tracks (Fig. 3g; Detour post vs. pre P = 0.0011; Linear tracks post vs. pre P = 0.0703; Wilcoxon signed rank test). With a two-parameter measure, detour showed significant plasticity in three pixels within the significant region, while non-detoured linear

tracks showed significant plasticity in one pixel within the significant region (Fig. 3h).

## Impact of detour on representations during future run sessions was predictable from sleep

Since we observed that detour-induced plasticity was expressed later during offline states, we asked whether this detour-induced plasticity was long-lasting and could also impact the activity during the following reversal run. Indeed, we found that neuronal activity changes (i.e., remapping) induced by the unexpected detour experience were only partially restored during the reversal run (Fig. 4a), which we interpreted as a form of representational drift. We defined the detour-induced neuronal changes that reversed during the reversal run as 'elastic' while the persisting changes as 'plastic' (see "Methods"). We were particularly interested in the additional network plasticity/elasticity caused by the detour experience, and thus we quantified them by comparing detoured tracks with non-detoured tracks. We expected that detour-induced elastic changes would result in a higher similarity between the circuit patterns before and after the detour

compared to before and during the detour. Meanwhile, detour-induced plastic changes would result in larger circuit changes between before and after detour than when there was no direct detour impact (Fig. 4b). We tested these hypotheses by studying hippocampal network changes across run sessions at the single neuron level and neuronal ensemble level. We categorized run session pairs into four groups based on how they were related to detour: (1) session pairs for tracks without direct detour (T1 & T3); (2) before detour versus detour session pairs; (3) after detour versus detour pairs; (4) before detour versus after detour pairs.

At the single neuron level, we studied the spatial tuning curve similarity on the first and the last stationary 50 cm linear segments of the detoured or non-detoured tracks across sessions, which match the vector length of tuning curves across sessions (Fig. 4c). The middle track segments were excluded as there was no direct correspondence between the 150 cm U-shape detour segment and the 50 cm mobile segment (Fig. 1d). Both plastic and elastic place maps were observed in relation to detour (Fig. 4d), with detoured tracks having significantly higher ratio of plastic and elastic place maps than non-detoured T1 and T3 (Supplementary Fig. 13; detoured tracks, plastic maps $21.00 \pm 2.45\%$; elastic maps $31.80 \pm 3.03\%$; non-detoured tracks, plastic maps $13.73 \pm 2.28\%$; elastic maps $2.92 \pm 1.04\%$; mean±s.e.m.). By definition, the plastic/elastic cells belonged to a subpopulation undergoing strong remapping from the pre-detour to the detour session. T1 and T3 tracks had fewer elastic place maps because cells generally exhibited stable spatial tuning across sessions, which reduced their plastic/elastic indexes to around 0, rendering them neither plastic nor elastic. Across session pairs, the place map similarity was highest between tracks that were not directly detoured (group 1), strongest remapping occurred between before detour and detour run sessions (group 2), while the other two groups (3 and 4) exhibited intermediate levels of remapping (Fig. 4e; $P$-values of ANOVA multiple comparison over the dimension of relationship to detour: NoDet and BeforeVsDet $3.8 \times 10^{-9}$; NoDet and AfterVsDet $3.8 \times 10^{-9}$; NoDet and BeforeVsAfter 0.0175; BeforeVsDet and AfterVsDet 0.0087; BeforeVsDet and BeforeVsAfter $3.8 \times 10^{-9}$; Degree of freedom 2641). This indicates the novel detour differentially impacted the associated T2/T4 tracks stronger compared to the non-detoured T1/T3 tracks, suggesting its experience was different than a simple exposure to a novel track environment. We also found that cells that were more active during the detour session compared with the pre-detour session were more plastic, while cells that were more active on the pre-detour session were more elastic (Supplementary Fig. 14b). Based on a multivariate linear regression model, both pre-detour and detour tuning curves significantly and best explained post-detour tuning curves (Fig. 4f), which revealed the intricate external stimuli-driven and internal experience-driven features of hippocampal activity.

At the neuronal pair level, the relative order as well as the time lag between neurons can be characterized by spike-time CCGs. The CCG can be studied at a behavioral timescale, which reflects the place maps sequence on the track, or at a theta timescale, which reflects the sequential neuronal order during compressed temporal coding[25]. Unlike the place map correlation analysis, the CCG analysis was conducted in the time domain and was amenable to comparisons across different track lengths. As a result, we compared the spiking activities on the entire tracks across sessions. We showed that CCGs at the behavioral and compressed theta timescales were impacted by the detour experience (Supplementary Fig. 15a). To quantify the impact of detour on CCGs from before to during and after detour sessions, we measured the cosine similarity of low-pass filtered CCG curves across sessions. At both timescales, the CCGs changed the least when there was no direct detour impact, and changed the most between before and during detour run sessions (Supplementary Fig. 15b; $P$-values of ANOVA multiple comparison over the dimension of relationship to detour: Behavioral scale: NoDet and BeforeVsDet $3.8 \times 10^{-9}$; NoDet and AfterVsDet $1.4 \times 10^{-4}$; NoDet and PreVsPost $8.0 \times 10^{-5}$; BeforeVsDet and

AfterVsDet 0.1289; BeforeVsDet and PreVsPost 0.0066; Theta scale: NoDet and BeforeVsDet $3.8 \times 10^{-9}$; NoDet and AfterVsDet $1.6 \times 10^{-4}$; NoDet and PreVsPost $5.6 \times 10^{-8}$; BeforeVsDet and AfterVsDet 0.0881; BeforeVsDet and PreVsPost 0.0245).

At the neuronal ensemble level, we studied how cells were recruited into cell assemblies to co-represent space[32,39,58,59] (see "Methods"). We detected cell assemblies for each run direction, track, and session. In example cell assemblies, their cellular patterns illustrated cells' contributions to the assembly, and their temporal patterns showed assemblies' activations over time (Supplementary Fig. 15c). We investigated detour's impact on cell assemblies by comparing the similarity of significant cell assemblies' cellular patterns across run sessions. We found the detour experience changed the assemblies from before to during and after the detour experience significantly stronger compared to not-detoured tracks (Supplementary Fig. 15d; $P$-values of ANOVA multiple comparison over the dimension of relationship to detour: NoDet and BeforeVsDet $3.8 \times 10^{-9}$; NoDet and AfterVsDet $3.8 \times 10^{-9}$; NoDet and PreVsPost $2.2 \times 10^{-4}$; BeforeVsDet and AfterVsDet 0.0937; BeforeVsDet and PreVsPost $2.2 \times 10^{-7}$). We also found that active pre-detour assemblies were more elastic while detour-active assemblies were more plastic in the post-detour reversal run (Supplementary Fig. 14e).

To further investigate whether the plastic impact of detour on the post-detour reversal run was related to its consolidation and network reconfiguration expressed during the preceding offline states[18,31,60], place map sequences during Run4 were split into drift and stable cell sequences based on their place map correlation between Run1 and Run4. We built a Markov model of activity in Sleep3, which was post-detour experience but before the reversal run, to predict the drift or stable sequences in Run4, and compared that with the Markov model built from Sleep1. We found that Sleep3 better explained the Run4 drift sequence than Sleep1, while this difference was not observed for the Run4 stable sequence (Fig. 4g–j; Normalized probability: Stable sequence Sleep1 vs. Sleep3 prediction: $P = 0.2553$; Drift sequence Sleep1 vs. Sleep3 prediction: $P = 0.0113$; Probability percentile vs. Shuffle sequence: Stable sequence Sleep1 vs. Sleep3 prediction: $P = 0.5417$; Drift sequence Sleep1 vs. Sleep3 prediction: $P = 0.0072$; Wilcoxon signed rank test). A similar phenomenon was observed when we used sleep to predict sequential activities of drifting cells in theta cycles (normalized probability: Stable theta sequence Sleep1 vs. Sleep3 prediction: $P = 0.0703$; Drift theta sequence Sleep1 vs. Sleep3 prediction: $P = 0.0012$; probability percentile vs. shuffle sequence: Stable theta sequence Sleep1 vs. Sleep3 prediction: $P = 0.0174$; Drift theta sequence Sleep1 vs. Sleep3 prediction: $P = 0.0052$; Wilcoxon signed rank test; Supplementary Fig. 16). This further demonstrates that dynamic hippocampal internal network patterns expressed during sleep are predictive over a variety of changes expressed in future network-level context representation that include representational drift and remapping at both behavioral and compressed time scales. While stable sequences across detours were compatible with network pre-configuration throughout the task, the more recent detour experience-induced plastic changes were consolidated during the offline post-detour sleep and were correlated with the future representational drift.

## Flickering representations of alternate environments

We found a significant activation of detour assemblies during the post-detour run session (Supplementary Fig. 14h), which suggested that even when the animals were immersed in a certain environment, spatial representation of alternative environments could be activated in the form of a competing alternate cognitive map[1,2]. To explore this parallel representation during the awake exploratory run state, we concatenated place maps from the related track segments and decoded the spike activity during detour laps at a theta timescale (40 ms time bin). During the detour session, there were instances with high

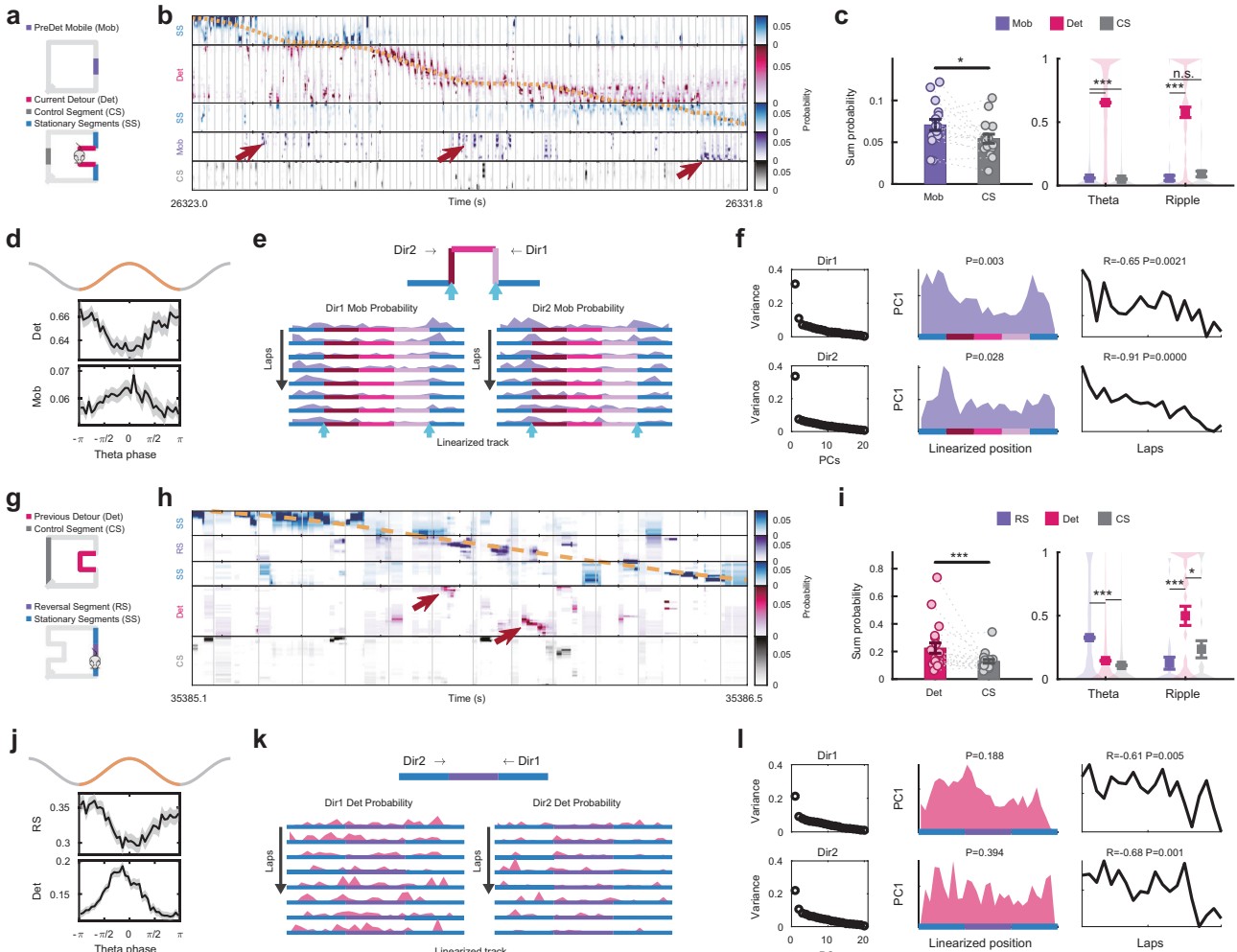

**Fig. 5 | Spontaneous flicker between alternate spatial representations during run. a** Configuration of context decoding during laps on the detoured track. **b** Example showing decoding results at fine time scale (40 ms) with animal's trajectory (orange dashed line). Vertical gray lines mark theta peaks. Instances with high probability decoding of the pre-detour mobile (non-local) segment during the detour session are marked with arrows. **c** Left, decoding probability of the pre-detour mobile segment compared with the control segment during all laps. Right, decoding probability of detour, mobile, and control segments during theta and ripple states. Wilcoxon signed rank test. **d** Theta phase modulation of decoding probabilities for detour and the pre-detour mobile segment. **e** Spatial-temporal pattern of decoded probability of pre-detour mobile segment during detour run. High probabilities around entering or leaving detour corners are marked by blue arrows. **f** Principal component (PC) analysis of the pre-detour mobile segment probability for direction 1 (top) and direction 2 (bottom) laps. PC1 was the dominant component for both directions. In PC1, probabilities were higher around

entering or leaving detour corners and decayed as a function of the number of laps. **g, h** Configuration of decoding (**g**) and an example (**h**) during laps on the post-detour reversal track during the post-detour session. **i** Decoding probability of detour segment (non-local) compared with control segment across behavioral states during post-detour run. Wilcoxon signed rank test. **j, k** Theta phase modulation (**j**) and spatial-temporal pattern of detour probability during post-detour run (**k**). **l** PC analysis of detour probability for direction 1 (top) and direction 2 (bottom) laps. PC1 was the dominant component for both directions. In PC1, probabilities had a relatively uniform distribution and decayed as a function of the number of laps. In panels (**c**–**f**) and (**i**–**l**), results were obtained by concatenating data from all animals and detour sessions. Lines and shaded areas in (**d**) and (**j**) indicate mean and s.e.m. Data in (**c**) and (**i**) are displayed as mean ± s.e.m. with each dot representing one animal, direction, and detour session ($n = 20$). ***$P < 0.001$, **$P < 0.01$, *$P < 0.05$, n.s. = not significant. Source data are provided as a Source Data file.

decoding probability of the 50 cm pre-detour mobile segment (Fig. 5a, b), which was significantly higher than the probability of the equal length (50 cm) control segment on the non-detoured parallel track (Fig. 5b, c; $P = 0.0296$; Wilcoxon one-sided signed rank test with pre-detour mobile segment having higher probability). This decoded probability was also stronger than those of different control segments (Supplementary Fig. 17a, b) and was not caused by a similarity of place maps between detour and the pre-detour mobile segments (Fig. 1f).

We found the pre-detour mobile segment had significantly stronger representation than control segments only during the theta oscillation (Theta state $P = 1.2 \times 10^{-11}$; Ripple state $P = 0.7050$; Wilcoxon signed rank test) while the novel detour representation was dominant over pre-detour and control segments during both theta and ripple

oscillation brain states (Fig. 5c). We next investigated how the representations of pre-detour and detour were related to the phase of theta and what were their spatial-temporal patterns. The representations of detour and pre-detour mobile segments were modulated by the phase of theta in dorsal CA1 pyramidal layer, with detour having higher decoding probability around theta troughs while pre-detour mobile segment around theta peaks (Fig. 5d). To alleviate a potential impact from decoding noise, we defined strong representation epochs as the time bins with at least 3 active cells and decoding probability higher than 0.9 for a given segment. Under this stricter regime, the pre-detour mobile segment had stronger representation epochs than the control segment (Supplementary Fig. 18a; $P = 0.0273$; Wilcoxon one-sided signed rank test with pre-detour mobile segment having stronger

representations). The strong representation epochs of the pre-detour mobile segment had significantly biased distribution near theta peaks, while the strong representation of the control segment had a more uniform theta phase distribution (Supplementary Fig. 18c). In terms of spatial-temporal activation patterns, our animals expressed strongest pre-detour mobile segment representation around entering or leaving the detour boundary corners during the early laps (Fig. 5e). Based on principal component (PC) analysis, the spatial-temporal pattern of this representation was dominated by the first PC displaying the highest probability around the detour boundary corners and decaying across laps (Fig. 5f).

During post-detour reversal run, we observed instances of high decoding probability for the detour segment (Fig. 5g, h), with average detour decoding probability and the frequency of strong representation epochs significantly higher than the control segments (Fig. 5i; $P = 8.0 \times 10^{-4}$; Supplementary Fig. 18b; $P = 0.003$; Wilcoxon one sided signed rank test with detour having stronger representations; Supplementary Fig. 17c, d). The representation of detour during post-detour reversal run was caused by the detour experience as it was not significant during the pre-detour run (Supplementary Fig. 17e, f; $P = 0.0793$; Supplementary Fig. 18b; $P = 0.5724$; Wilcoxon one-sided signed rank test with detour having stronger representations). The detour segment probability was significantly higher than controls during both theta state and ripple states (Theta state $P = 4.1 \times 10^{-24}$; Ripple state $P = 0.0379$; Wilcoxon signed rank test). During theta state, the reversal run had stronger representation during theta troughs while the detour segment (physically absent, recalled) during theta peaks, as measured by average probability or distribution of strong representation epochs (Fig. 5j). The theta phase distribution of strong representation epochs for the control segment was not significantly different from a uniform distribution (Supplementary Fig. 18d). The spatial-temporal pattern of detour representation (Fig. 5k) was dominated by the first PC. Although the detour probability spatial distribution was relatively uniform across the linear track, the decay of detour representation across laps during the post-detour run session was significant (Fig. 5l).

**Flickering representations were predictable from sleep and contributed to low-tuning cell activities**

Since the external stimuli of currently absent (non-local) contexts were not directly available to the animals, we inferred the flickering representation of the absent contexts primarily reflected the internal drive of hippocampal representations originated from experience. During both detour run and reversal run, we found that neuronal order dependencies obtained using the Markov model computed during the preceding sleep predicted more accurately the spiking activities within the flicker epochs representing the non-local context (previously explored) compared to the local context-representing epochs (Fig. 6a–f; Detour run: mobile-representing epochs, $n = 215$, vs. detour-representing epochs, $n = 5345$, normalized probability $P = 4.0 \times 10^{-8}$, percentile against shuffle $P = 0.0077$; Reversal run: reversal-representing epochs, $n = 1169$, vs. detour-representing epochs, $n = 384$, normalized probability $P = 4.5 \times 10^{-15}$, percentile against shuffle $P = 0.0339$; Wilcoxon rank-sum test). Thus, the flickering representations of the local and non-local contexts reveal the competition between the internally-driven recall and the combined external and internal drives on hippocampal current representation, segregated by hippocampal theta phases.

Since the flickering representation was strongly internally driven, we further hypothesized the flickering representation would result in spike activities not spatially-tuned to the current environment. During the post-detour reversal run, place cells exhibited activity preference between the current reversal and detour representation (Fig. 6g, h). We defined detour cells and reversal cells based on their spike preference to these representations, and found the detour cells had

significantly less stable spatial tuning within the post-detour session and experienced larger place map drift from pre-detour to post-detour session compared to reversal cells (Fig. 6i, j; $P = 0.005$, $P = 0.0436$; Wilcoxon signed rank test). This drift was different from the plastic place maps shown in Fig. 4d–f, where a post-detour place field was inherited from the detour session. Because the flickering representation of detour did not occur at a fixed position on the maze (Fig. 5l), the spikes of detour-representing cells did not show a significant spatial tuning anchored to the current context, but their activities could be explained by the flickering of a past detour experience.

## Discussion

We have shown that during a novel detour experience, the hippocampal CA1 network rapidly expressed novel time-compressed theta sequences that matched time-compressed sequence motifs active during preplay events in the preceding sleep session. The detour theta sequences were expressed several laps earlier than the first-time expression of waking rest replay of the detour experience, consistent with their emergence from native pre-existing sequence motifs and not experience-induced replay-driven time compression. Moreover, we demonstrated a dissociation between theta phase mechanisms (phase locking and phase precession of individual place cells) and theta sequence, indicating that single-cell temporal coding mechanisms were not sufficient to explain the neuronal ensemble grouping during theta sequence coding. Finally, we showed that a compelling aspect of the theta sequence, the rapid selection/grouping of different neurons together within a theta cycle, was already expressed during the preceding sleep as preplay and tuplet sequential motifs, and could not be explained by individual neuron properties like phase precession and locking.

Our observations on the order of expression of temporally-compressed neural sequences from preplay to theta sequences to replay are consistent with previous studies where (1) their emergence in this same order was found during postnatal neuro-development[41], (2) interrupting theta sequences during animal navigation resulted in impaired replay[42,61], and (3) lap emergence of theta sequences and replay in adulthood were investigated in isolation in separate studies, animals, and under different tasks[30,46]. To explain the rapid development of time-compressed theta sequences, we propose that their neural selection and allocation[56] from the preconfigured repertoire of sequence motifs rendered the rapid sequential coding of new environments. Several current findings support this view. First, we found significant preplay of the novel detour experience, including of animals' trajectory in the first 1-2 detour laps. Whereas previously, preplay was described only for linear track experiences, we found that preplay of detour trajectories crossed several detour corners, indicating the preconfigured structure could function as a template for sequential representation beyond linear spatial experiences. Second, we demonstrated that detour preplay had strong predictive power over early detour theta sequences, from cells' recruitment and allocation as place cells to their relative order during sequential representation of novel detours. Third, we found that sequential tuplet motif structures expressed during pre-detour sleep were significantly recruited and reused in early laps theta cycles. All these findings support the proposal that pre-configuration contributed to a relatively stable early-laps theta sequence representation, whose meaning could be further modified by association with specific external stimuli and generative extension via multiplexing into longer neuronal sequences[21] and richer cell assemblies[32] during the novel experience.

The novel detour experience caused strong place cell remapping and gave rise to network-level plasticity. The novel experience-induced plasticity was consolidated during the following offline brain states and was expressed during post-detour online brain states, when it impacted the spatial representation of the unchanged linear environment during the post-detour reversal run. This altered

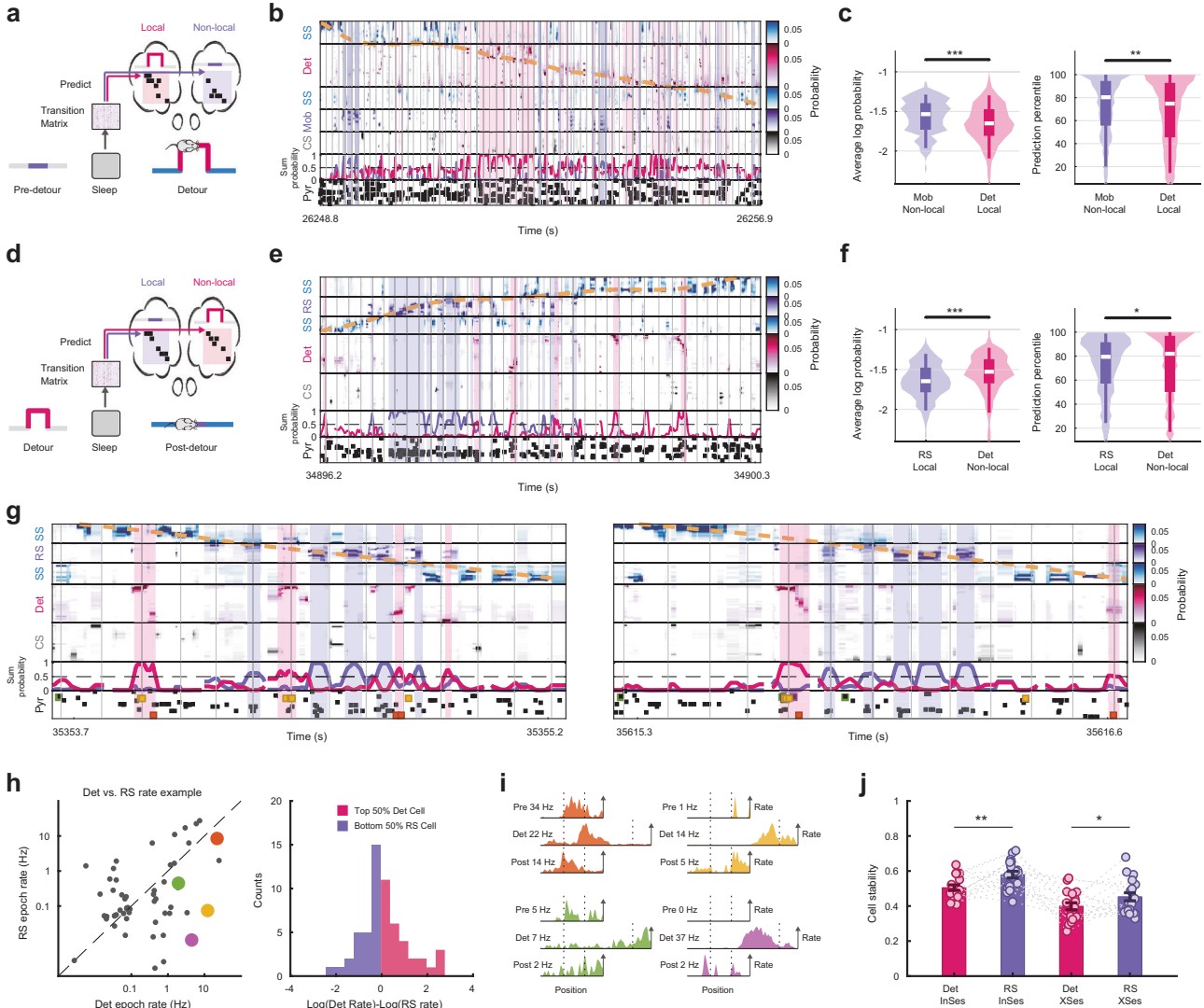

**Fig. 6 | Neural activities during flickering were predictable from sleep and decreased tuning stability. a** Using the pre-detour sleep probability transition matrix to predict spike activities during detour run flickering representations. **b** Example showing decoding result at fine time scale during detour run. Vertical gray lines mark theta peaks. Purple shaded areas are Mob-representing epochs (non-local) and red shaded areas are Det-representing epochs (local) during active run (>10 cm/s). **c** Probabilities of spiking activities predicted by pre-run sleep during Mob- and Det-representing epochs (Left) and probability percentiles compared with shuffle sequences (Right). Wilcoxon rank-sum test. (*n* = 215 for mobile-representing epochs; *n* = 5345 for detour-representing epochs). **d** Using pre-reversal sleep probability transition matrix to predict spike activities during post-detour reversal run flickering representations. **e** Example showing decoding result at fine time scale during post-detour reversal run (shaded area highlighted local/non-local representing epochs). **f** Probabilities of spike activities predicted by pre-run sleep during RS- and Det-representing epochs (Left) and probability percentile compared with shuffle sequences (Right). Wilcoxon rank-sum test. (*n* = 1169 for

reversal-representing epochs; *n* = 384 for detour-representing epochs). **g** Examples showing decoding result at fine time scale during post-detour run. The activities of 4 example pyramidal cells are highlighted. **h** Left, example showing pyramidal cells activities during detour-representing epochs and RS-representing epochs. Right, example showing distribution of detour-representing and RS-representing cells. **i** Spatial tuning curves of 4 highlighted example cells in (**g**) and (**h**) during pre-detour, detour, and post-detour sessions. Peak firing rates are marked for all the sessions. **j** Within session (post-detour) across-laps place map stability and across-sessions (pre-detour vs. post-detour) place map stability of detour- and RS-representing cells. Wilcoxon signed rank test. Bar plots in (**j**) display mean ± s.e.m. with each dot representing one animal, direction, and detour session (*n* = 20). Violin and box plots in (**c**) and (**f**) displayed the data distribution with whiskers representing the 5th to 95th percentile range; the box represented the 25th to 75th percentiles, and the center dot indicated the median value. ***P < 0.001, **P < 0.01, *P < 0.05. Source data are provided as a Source Data file.

representation of the unchanged external context acted as a representational drift. The hippocampal representational drift has been studied with both calcium-imaging[11,15,16] and electrophysiological methods[62,63]. Here, we built a conceptual model to partially explain the within-day representational drift, where the in-between experience could contribute to the altered representation. In addition to a previous study where the interleaving neuronal activity was monitored between exposures[64], our study linked the altered representation with a specific detour experience that we introduced. We propose that, as animals continuously engage in

unaccounted active exploration during and outside the recording, any event happening during this time could be treated by the animals as task-relevant, and some of those can go beyond the modality that we monitored. Those unmonitored events could update the internal neuronal network state and thus alter the representation of the unchanged external environment. This interpretation is consistent with recent studies showing the contribution of active experience to the representational drift[16,17], and a previous study exemplifying carry-over effects of geometric changes to the same environment on CA1 place cell activity[65]. In our study, we critically

linked this experience-induced representational drift with the consolidation process occurring during the interleaving sleep.

Our findings identified one expression of this experience-induced plasticity in hippocampal CA1 during the flickering representation of an alternate context. We found this flickering representation was strongly contributed by intrinsic network dynamics predictable from sleep, and was supported by spiking activity at CA1 theta peaks, while representation of the current context was conveyed by spiking activity at CA1 theta troughs. A similar theta phase segregation is also found during hippocampal theta sequences when activities at theta peaks represent past or future locations, while those at theta troughs the current animal location[25,41]. This indicates a more general principle by which prospective (e.g., imagining) and retrospective (e.g., recall) coding preferentially occurs around theta peaks while current context is represented around theta troughs, possibly enabling generative temporal binding of current and recalled/imagined representations within a theta cycle timescale. These differential activities may be contributed by the upstream CA3 area, where neural activity patterns also flicker between representations of two contexts during[66] and after artificially-evoking context-specific external cues[67] or when rats were presented with a choice/decision between two current, incoming maze trajectories[68–70]. Here, we specifically investigated whether the CA1 area could spontaneously represent two environments in parallel, one being physically available and currently explored (i.e., local) and one being unavailable and retrieved from memory (e.g., a previous detour, non-local). Importantly, as this flickering representation didn't occur at fixed spatial locations, the spike activities during the flickering exhibited weak spatial tuning with respect to the current context. These low-tuning cells and their associated alternate spatial co-representations were likely omitted by previous studies using rigorous place cell inclusion criteria. However, this flickering-induced representational drift may not be a deficit of hippocampal representation, but rather indicates higher-level cognitive functions beyond pure coding of the present, local context[56].

Our study characterized the interplay between internally- and externally-driven brain states during hippocampal representation of spatial detour experience. First, internal states could facilitate rapid encoding and representation of novel environments via rapid recruitment and incorporation of preconfigured preplay and tuplet sequences into corresponding novel detour theta sequences. Second, the novel detour experience was consolidated during sleep and modified the internal state through network-level plasticity. Last, the retuning/reconfiguration of internal state activity continuously sculpted the hippocampal spatial representation during re-exposure to the unchanged linear environment. This enabled flickering between a present representation and the one of the recalled, currently absent, context, resulting in an updated representation manifested as representational drift. These forms of internally generated activity, which include look-ahead theta sequences, experience-induced representational drift, and context-dependent flickering, constitute non-local coding, whereby the hippocampus transiently expresses representations not anchored to the animal's present location or ongoing sensory experience. Our results highlight the dynamic balance between the intrinsic stable and flexible frameworks, and how those frameworks incorporate the external input while shaping and updating a cognitive map.

## Methods

### Animals

Five Long-Evans adult male rats with body weight ~350 g were used for this study. Animals were housed on a normal circadian rhythm (Light: 9 a.m. to 9 p.m.; Dark: 9 p.m. to 9 a.m.). The experiments started at the light phase of the circadian rhythm to facilitate adequate sleep. Animal handling and experimental procedures were approved by the IACUC at Yale University and were performed in agreement with the NIH guidelines for ethical treatment of animals.

### Surgery and experimental design

Animals were implanted bilaterally with 32 independently movable tetrodes (Rats 3–5) or two independently movable 64-channel 8-shank silicon octatrodes (Neuronexus probes, Rats 1-2) under isoflurane anesthesia. Craniotomy was performed above area CA1 of the hippocampus (centered at 4 mm post-bregma, 2 mm lateral to midline). The reference electrode was implanted posterior to lambda over the cerebellum. During the following several weeks post recovery, the tetrodes and silicon probes were advanced daily while animals rested in a high-wall opaque sleeping box ($30 \times 45 \times 40$ (h) cm).

The experimental apparatus was a $150 \times 150$ cm rectangular elevated linear track maze (tracks 1-4, T1 to T4) with two additional parallel and two orthogonal tracks inside the square. All tracks were 150 cm long, 6.25 cm wide and 75 cm above the floor. Before the detour task day, the animals had explored all eight 150 cm linear tracks end-to-end and had access to the connected rectangular-shaped outer tracks, while access to the inner tracks was blocked by 20 cm-high, 10 cm-wide barriers. A permanent barrier was placed throughout this experiment and the previous days between tracks 1 and 4. Animals explored the tracks for chocolate sprinkle rewards placed on both sides of the permanent barrier at the adjacent ends of tracks 1 and 4. During the detour task day, animals first had a sleep session (Sleep1) where they were placed in the familiar sleep box for ~2 h. After that, while rats were still in the opaque sleep box, the maze was brought into the room and installed. Subsequently, run session 1 (Run1) began when the animals were transferred onto track 1 next to the permanent barrier and explored track 1 in both directions for at least 3 laps while access to any other track was being blocked. Next, track 1 barrier adjacent to track 2 was lifted, and the animals could explore tracks 1-2 to collect food rewards placed at the end of this L-shape portion of the maze. This barrier-lifting procedure was repeated two more times after 1-2 laps for tracks 3 and 4, until the animal could explore all four outer tracks for at least 10 laps for rewards always placed at the 2 ends of the maze.

Run session 1 lasted around 30 min and after that, animals were placed back in the sleep box for ~2 h (Sleep2). During this sleep session, an unexpected first detour was introduced on either track 2 or track 4, counterbalanced across different animals. Specifically, the 50 cm middle segment of the detoured track was removed (therefore called mobile segment), and two barriers were lifted to give the animals access to a 150 cm-long U-shape detour segment connecting the first and third 50 cm stationary segments of the respective detoured track, one at a time, as illustrated in Fig. 1a. After sleep session 2, animals were placed on the maze to start the detour run session (Run2). The entire 700 cm maze was explored on both directions for at least 10 laps for food rewards. After completing run session 2, the animals were briefly blocked on track 1 or track 3 by barriers while the experimenter reversed the first detoured track and introduced a novel, analogous detour in the parallel outer track. Then, the temporary blocking barriers were lifted, and animals started to explore the 700 cm maze with the new detour configuration for at least 10 full-maze laps (run session 3, Run3). After Run3, the animals experienced a post-detour sleep session in the sleep box (Sleep3), while the experimenter restored the maze to the original pre-detour configuration as in Run1. After Sleep3, the animals explored again the original maze configuration where they could access all four outer tracks (Run4); the middle segments of the linear tracks that were previously detoured were equivalent to a reversal segment. Run4 was followed by one last sleep session in the sleep box (Sleep4) that ended the day of the detour experiment.

### Detour session assignment

In Rat 1 and Rat 5, track 2 was detoured in run session 2, while track 4 was detoured in run session 3. In the other three animals, track 4 was detoured in session 2, and track 2 was detoured in run session 3. In

some analyses where track 1 and track 3 were used as controls to compare against track 2 and track 4 changes, virtual detour sessions were defined for track 1 and track 3. In those analyses, track 1 had the same pre-detour, detour and post-detour sessions as track 2, and track 3 had the same pre-detour, detour and post-detour sessions as track 4.

## Electrophysiology data acquisition

Electrophysiological data were collected using a 128-channel digital Neuralynx data acquisition system (DigiLynx) with Cheetah software. Raw signals were recorded at 30 kHz and were band-pass filtered between 1 and 6000 Hz. Spikes were obtained by high-pass filtering the raw signal above 600 Hz and triggering signal acquisition by passing a 50 μV threshold. Single cells were isolated offline using the manual clustering method Xclust3 as described before[20]. Putative pyramidal cells were distinguished from interneurons based on inter-spike intervals, average rate, and waveforms[20]. To eliminate the potential impact of electrode drift, only clusters with stable amplitudes across all sessions were kept for further analysis. We collected 61, 27, 57, 11 and 45 stable putative pyramidal cells across all run and sleep sessions from our 5 animals. After the completion of all experiments, all rats were perfused intracardially with 10% formalin, and their brains were fixed, sectioned, and stained using Cresyl violet to reconstruct all electrode tracks.

## Behavior data processing

The animal's position was monitored via headstage LEDs and an overhead camera with a sampling frequency of 30 Hz. Animals' 2D position was collapsed to a linear position where the range of each track and the linear position of detour corners were marked. Direction 1 was defined as clockwise movement in the square maze, while direction 2 was defined as counter-clockwise movement. Hippocampal activities were analyzed relative to individual tracks and moving direction.

## Place maps on tracks

Place maps were computed as the ratio between the number of spikes and the time spent in 2 cm spatial bins along the track, smoothed by a Gaussian kernel with a standard deviation of 2 cm[20]. Bins where the animal spent a total of less than 0.1 s and periods during which the animal's velocity was below 10 cm/s were excluded from place map computation.

## Spatial information and within-session stability

Spatial information (SI; bits/spike) quantifies the nonuniformity of cells' firing rates over spatial bins and was computed as described before[26]. To quantify the within session place map stability for each cell, laps were randomly split into two groups, the averaged place maps were obtained for each group, and the cosine similarity between two place maps was computed. This process was repeated 100 times, and the averaged cosine similarity was defined as within session place map similarity.

## Remapping between mobile middle segment of the detoured tracks and the detour segment

Spatial remapping between the pre-detour mobile segment (50 cm) and the detour segment (150 cm) was assessed at single-cell and population levels.

At the single cell place map level, we first compared the place map similarity between the pre-detour mobile segment and the detour segments. As the detour segment was 3 times longer, they were divided into three 50 cm segments separated by detour corners, and each segment was compared to the pre-detour mobile segment. We also added two additional configurations to test different hypotheses. First, we compared the entire 150 cm detour segment with the 50 cm pre-detour mobile segment to check if the detour segment was a stretched

version of the pre-detour mobile segment. Because the segment lengths were different, we used 6 cm spatial bin resolution for the 150 cm detour segment and 2 cm spatial bin resolution for the 50 cm pre-detour mobile segment, so we can match the total number of spatial bins. Second, we compared the entire 150 cm detour segment to the entire 150 cm pre-detour linear track. The Pearson correlation between the place maps of the same cell was computed, where we allowed segments to shift up to 2 bins (up to 4 cm) to reach the maximum correlation value. The results were compared with surrogate datasets with cell ID shuffle ($n = 1000$), where the correlations were computed in the same manner. Note that within each segment pair, only cells with peak firing rate above 2 Hz on at least one segment were included in this analysis.

We compared the firing rate similarity on the pre-detour mobile segment and the detour segment to determine if cells that were more active on the mobile segment would also be recruited to represent the detour segment. We compared the similarity against other middle segments during the pre-detour session, which were middle segments from T1, T3, and the opposite track. We measured how many cells that had place fields on the detour segment also had place fields on the pre-detour middle segments. We also directly correlated the mean firing rates on the detour segment and the pre-detour middle segments. To better visualize the distribution, we sorted cells based on their mean firing rates on the pre-detour middle segments, binned the percentile, and computed within each percentile bin for those cells the mean and s.e.m. activity percentile on the detour segments.

At the population level, the population vector (PV) cosine similarity between spatial bin pairs was computed. The PV was defined as the vector of all cells' firing rates at a given spatial bin. As this was a bin-by-bin analysis, there was no need to match segment lengths. The PV similarities of the spatial bin pairs within the pre-detour mobile segment and the detour segment region were compared with those of bin pairs within the other tracks' middle segment and detour segment regions.

## Detection of frames during sleep and waking rest

We defined short epochs with strong multi-unit activities as frames. Frames were detected during non-REM sleep and waking rest immobility periods where the velocity of the animal was smaller than 2 cm/s in the sleep box or on the maze. During these periods, we binned population activity of all putative pyramidal neurons with 1 ms time bin and smoothed the activity by a Gaussian kernel with a standard deviation of 15 ms. Periods when the population activity exceeded 2 standard deviations above the mean were considered as frame candidates, and then the frame durations were extended to when the population activities exceeded the mean value. Frame candidates which lasted between 100 ms and 1200 ms and contained at least 5 distinct active pyramidal neurons were considered as frames and used in further analysis.

## Bayesian decoding of location and trajectories

To decode location and trajectories based on neural activity, we employed a memoryless Bayesian decoding algorithm either with clustered or cluster-less spikes[50,52,71]. A small value ($mean\ rate \times 10^{-5}$) was added to the average firing rate to reduce instances of zero decoding probability. The decoding time bin was 20 ms in sleep frame decoding, waking rest frame decoding, and theta sequence decoding. The decoding time bin was 40 ms to analyze representations of alternative detour or middle segments during the run. The decoding time bin was 200 ms to validate the Bayesian decoder and compute the decoding error during the run.

To test whether the spike pattern specifically represented a track, we included several tracks in the place map template, e.g., four tracks in four sessions in one direction (16 tracks in total), and the decoded probability was normalized across these tracks. We called this the

concatenated place map decoding, where tracks in different sessions competed for decoding probabilities. When not specified, place maps on the four tracks in four sessions in one direction were included during waking rest and sleep decoding.

## Weighted correlations and maximum jump distance

To determine whether the decoded result of a frame represented a sequential trajectory, a linear correlation between time and location was computed, weighted by the associated posterior probabilities. To study if the decoded trajectory exhibited large jumps (discontinuities in trajectory sequences) within the frames, the peak decoded virtual location was computed for each temporal bin. The maximum value of the differences between peak decoded locations in consecutive bins was defined as the maximum jump distance for a particular frame. The maximum jump was then normalized by the length of the track.

## Theta phase detection

In order to identify theta oscillation phases and cycles, an optimal recorded channel was determined based on the number of detected pyramidal cells. Then the LFP was band-pass filtered into the theta range (7–10 Hz), and a Hilbert transform was conducted on the filtered trace to obtain the theta phase information. By default, the interval between consecutive theta phase peaks was defined as a theta cycle.

## Theta phase locking and phase precession detection

To construct the sequence based on the preferred theta phase, spikes within the first two detour laps on the detour segment were first isolated. Cells with significant theta phase modulations were detected with Rayleigh Z test and their preferred spike phases were defined as the peak phase in the spike histogram with bin size of 20 degrees. Significantly phase modulated cells were sorted based on their preferred spike phases and compared with the actual place map sequence (directional sensitive) on detour segment with Spearman's rank-order correlation.

To test the significance of theta phase precession in the first two detour laps, place fields on the detour segments were identified. Theta phase, as well as normalized position in the place fields, were extracted from all the spikes in the first two detour laps on detour segments. The Pearson correlation coefficient and $p$-value between theta phase and normalized position were computed.

## Theta sequences detection

First, theta cycles occurring during active behaviors (velocity > 10 cm/s) were isolated, and cycles with more than 3 neurons were used for further analysis. Bayesian decoding was applied, including all the spatial bins in the current maze, with a decoding window of 20 ms, and was performed for each theta cycle. The spatial bin was then corrected based on the current location of the animal (within ± 40 cm) to reveal the differences between actual and predicted locations. The quadrant ratio was computed by summing the probabilities in quadrants 1 (immediate future) and 3 (recent past) relative to the current location of the animal and subtracting the probabilities in the other two quadrants (2 and 4) from their sum (directional sensitive). This difference was then normalized by the sum of all probabilities in quadrants 1–4 as used previously[41]. To test the significance of the theta sequence, a Wilcoxon signed rank test was conducted to test whether the distribution of quadrant ratios of the targeting theta cycles was significantly positive.

## Cross-correlograms (CCGs) between cell pairs

Spike activities of cell pairs on a given track, session and run direction or during sleep frames were extracted to compute CCGs. We computed the center of mass from CCG as the temporal bias across cell pairs. At the compressed time scale, we used the CCG within a ± 200 ms window around 0 to compute the bias[53]. At the behavioral time scale, we used the CCG within a ± 1000 ms window around 0 to compute the bias. The temporal bias can be correlated across time scales or correlated with place field distances across different tracks and directions. To get the temporal bias during sleep, we used spiking activities during sleep frames (all sleep frames or selected preplay/replay frames) to compute CCG. The temporal bias was defined as the center of mass within the ± 200 ms window around 0. To correlate temporal bias with place map bias on the track, we first found cell pairs with overlapped place fields on the given track segment. Then we computed the place map bias, defined as the difference of the center of mass from the partially overlapping place fields.

To study how the CCG temporal structure changes across run sessions, we obtained the behavioral time scale CCG curve (low-pass filtered below 0.5 Hz, within ± 1000 ms around 0) and theta time scale CCG curve (low-pass filtered below 30 Hz, within ± 50 ms around 0). The cosine similarity of those CCG curves was measured across run sessions given the CCG curves had significant peaks (> mean+ 2 SD & > 2 counts).

## Spike jittering for theta phase precession and theta sequence

To test the dissociation between theta phase precession and theta sequence, we introduced temporal jittering to spike timing. We defined a jittering temporal range stepping from ± 5 ms to ± 75 ms. For a given jittering temporal range, each spike timing was independently shifted by a random value drawn from the uniform distribution centered at zero and with a minimum at the negative and maximum at the positive of the jittering temporal range (e.g., − 10 ms to 10 ms for the ± 10 ms jittering). The jittered spikes were generated once, and the same dataset was used for theta phase precession and theta sequence detection with the methods mentioned above.

## Comparison between theta sequence and waking rest replay significance

Despite the observation of expression of detour theta sequences in the early laps before the emergence of detour replay, there were several challenges in directly comparing their occurrence time and lap. The major difficulty was that the theta sequence and replay were detected with different methods. The detection of theta sequence emphasized the sweep from the past to the future in a theta cycle, whereas the detection of replay focused on finding a continuous decoding trajectory correlated in time and space. Furthermore, there were more theta cycles than spiking frames where replay could occur, which would make the theta sequence more likely to be detected due to more tests. Here, several efforts were made to make the occurrence of the detour theta sequence and detour replay technically comparable.

First, since theta sequence and replay were detected with different methods, we ensured that both detection methods had similar false positive rates, or the detection of theta sequence would have a lower false positive rate than replay. We used false positive rates rather than false negative rates because there was no ground truth in determining replay or theta sequence, while the false positive rates could be accessed from the temporal shuffle datasets. To achieve that, we first decided the significant criteria for replay detection. In the result section, we used the absolute weighted correlation larger than 0.6 and the normalized maximum jump smaller than 0.4 as used previously[32,51], and tested a combinatorial range of these parameter values in Supplementary Fig. 6. For the main parameters, the false positive rate was estimated by checking how many representations passed the significant criteria within the temporal shuffle datasets (Supplementary Fig. 6a). We found that 1.146% of the shuffle representations were detected as significant, thus we set a stricter detection criterion with a false positive rate (alpha value) of 1% for theta sequence detection. To make sure the actual false positive rate of theta sequence detection was lower than replay detection, we conducted a time bin shuffle for all the theta cycles within the first two laps, and randomly drew

1000 samples from the full population. For each sampled sub-population, we tested whether the distribution was significantly positive using a one-sided Wilcoxon signed rank with the corresponding alpha value. The procedure was repeated 1000 times, and we found the ratio of the significant subpopulation was not larger than the set alpha value (Supplementary Fig. 6).

Second, to make sure theta sequence and replay were detected at the same rate, for each frame that was tested for replay, we tested whether the whole population of theta cycles preceding that tested frame exhibited significant theta sequence. For each frame, we decoded all the theta cycles on the detour segment in a given direction before the frame's time point, and we computed quadrant ratios for all those theta cycles. Then, we tested whether the distribution of the quadrant ratios was significantly positive using a one-sided Wilcoxon signed rank test with significant level (alpha) smaller than the replay false positive rate (0.01 in Supplementary Fig. 2b, d).

With the above method, the time and lap when a detour replay was first detected was defined as the time and the lap of the significant replay. With the corresponding false positive rates, the time and the lap when the distribution of quadrant ratios of theta cycles first showed significance was defined as theta sequence significant time and lap.

## Parameters estimation for Markov chain model

The parameters for the probability model were estimated from sleep frames as described earlier[21]. Each sleep frame was represented by a sequence of cell IDs sorted by the center of mass of their spike times. The maximum likelihood estimation was used to estimate the parameters. Conditional probability in the transition matrix was given by:

$$\Pr(x_i|x_{i-k}\ldots x_{i-2}x_{i-1}) = \frac{n(x_{i-k}\ldots x_{i-2}x_{i-1}x_i)}{n(x_{i-k}\ldots x_{i-2}x_{i-1})} \quad (1)$$

where $x_i$ is the $i^{th}$ cell in the sleep sequence, $n$ is the count of sequence, and $k$ is the order of the model. In this study, we only used the 1st order mode,l and conditional probability Pr was represented as P2:

$$P2(x_i|x_{i-i}) = \frac{n(x_{i-1}x_i)}{n(x_{i-1})} \quad (2)$$

To alleviate the noise induced by low-firing cells, 0-value in the transition matrix was reset to the minimum non-zero value in the transition matrix, and 1 was reset to the maximum non-one value in the matrix. This operation was performed to avoid a 0-value in the estimated probability caused by the finite sample size of the real dataset, since during probability estimation of long sequences, one 0-value could make the probability of the whole sequence 0.

The unconditional probability P1 of each cell was given by:

$$P1(x_i) = \frac{n(x_i)}{N} \quad (3)$$

where $N$ is the length of sequential activity during sleep.

## Probability estimation of sequences from Markov model

For a given sequence x with length of n, the probability can be estimated based on the 1st order Markov model as:

$$\Pr(x) = P1(x_1)\prod_{i=2}^{n} P2(x_i|x_{i-i}) \quad (4)$$

This sequence probability is sensitive to the sequence length. To compare the sequence probabilities across different lengths, we introduced the normalized sequence probability as the geometry mean of the original probability. To evaluate the significance of a given sequence, the estimated probability was compared to shuffle datasets ($n = 500{,}000$), where random sequences with the same length were generated and their probabilities were computed. If the probability of the actual sequence exceeded the 95th percentile of the shuffled distributions, the sequence was considered significant.

## Tuplet extraction and analysis

Tuplets were defined as 2- and 3-cells sequential structures that were significantly repeated during sleep frames. The detection of tuplets followed the method described in ref. [21]. Briefly, 500 shuffle sleep frames were generated with spike activities selected by a weighted random sample without replacement based on the firing rates of cells in the original sleep. The shuffle sleep preserved the firing rates while it disrupted any cofiring or sequential structures during sleep frames. Then, in the original sleep frames, all the 2- and 3-cell sequential structures with at least two repeats were extracted. The number of repeats of those candidates were compared with the repeat times during shuffle sleep frames. If the repeating time was larger than 95% of the shuffle, then the 2-cell sequence was defined as a tuplet with a length of 2, and the 3-cell sequence was defined as a tuplet with a length of 3. Note that sequential structures with reversed orders can both be detected as tuplets if those cells are significantly co-active within the same frames.

To study the recruitment of sleep tuplets into forming early theta sequences, all the 2- and 3-cell sequential structures during the first 2 lap detour theta cycles were extracted, and among all these 2- or 3-cell sequential structures, we computed how many of them were tuplets expressed during pre-detour sleep. These values were compared against tuplets extracted from shuffle sleep.

## Pairwise order consistency between early lap detour theta cycles and detour preplay

To investigate the cell pair spike order consistency between early lap detour theta cycles and detour preplay frames, we first constructed the pairwise order probability matrix based on forward detour preplays. For a given cell pair A and B, we divided the number of frames where A fired before B (based on the center of mass of spikes within a frame) by the number of frames where both were active to get the order probability. The probability of B firing before A is 1 minus the probability that A fired before B. After getting this anti-symmetric pairwise order matrix, we examined all the pairwise spike orders in the early lap detour theta cycles. For each animal, direction, and detour session, we averaged the probabilities of those pairwise orders during theta cycles based on the probability matrix computed from sleep and subtracted the chance value 0.5 from the averaged probability. A positive result indicates the pairwise order is consistent between early lap detour theta cycles and forward detour preplays.

## Order correlation between spike activities in detour theta cycles and pre-detour sleep frames

To study the ensemble-level sequence order similarity between early lap detour theta cycles and pre-detour sleep frames, we computed the rank order correlation between sequences in theta cycles and sleep. Specifically, we first isolated the first two lap detour theta cycles with not less than 5 active cells, and computed the spike sequence order based on the center of mass of spikes for each theta cycle. Then we computed the spike sequence order in all the pre-detour sleep frames. Across all the theta-cycle and sleep frame pairs, we found pairs with not less than 5 common active cells to compute the Spearman rank order correlation. The at least 5 common active cells ensured there were at least 5! = 120 independent permutations. We compared the rank order correlation against 500 sequence permutation shuffles, and defined the pair as having significant correlation if the correlation value is higher than 95% of the shuffles. In the last step, the ratio of pairs with significant correlation (among

pairs with at least 5 common active cells) was compared against a 5% chance level.

## Comparing decoding probability between linear and detour tracks

To compare the decoding probability between linear and detour tracks during waking rest frames, concatenated place map decoding was used. Specifically, for a given direction, 6 place maps were included in the template: if the current session contained no detour (run 1 and run 4), we used 4 current linear tracks + 2 detour segments in detour sessions; if the current session contained detour (run 2 and run 3), we used 4 current tracks (1 detour segment + 3 linear tracks) + 1 detour segment in another detour session + 1 linear track for the current detoured track from pre-detour session. There are four groups of tracks: current linear tracks; detour segment in Run 2; detour segment in Run 3; pre-detour linear track if the current session is detoured (not shown in plots). The reason for using 6 tracks rather than 16 tracks from all the sessions was that place maps of linear tracks were relatively similar across sessions, and a replay of a specific linear track could be mirrored in several sessions. Including tracks from several sessions will dilute the decoding probability of linear tracks. Moreover, to alleviate the impact of firing rate difference on decoding probabilities, place maps from different tracks were rescaled to have the same mean firing rate.

## Impact of detour during run controlled by session pairs without direct detour impact

To assess the impact of detour during and after the detour session, we investigated how cells' place maps, cells' pair-wise spike order, and cell assemblies change across sessions. All the possible session pairs were characterized by two parameters. The first was the session difference between session pairs (ranging from 1 to 3), and another was how this session pair was related to detour. The impact of detour was assessed by matching session differences and comparing across groups. At each session lag, the difference across groups was obtained by Kruskal-Wallis ANOVA followed by post-hoc Tukey-Kramer multiple comparisons. To evaluate the overall difference between detour-related groups, a N-way ANOVA was built, where the model included session lag, animal ID and detour-related groups. Post-hoc multiple comparison test was performed over the dimension of detour-related groups with Tukey-Kramer correction. The place map correlation was only evaluated on the two 50 cm stationary segments to match the vector length across sessions. The CCG and assembly analyses were not impacted by different track lengths and were conducted on the whole track.

## Dimensionality reduction with UMAP

Dimensionality reduction method UMAP (UMAP: Uniform Manifold Approximation and Projection) was used to visualize the impact of detour on spike activities. Cells' place maps were computed for individual directions, tracks, sessions, and laps with a spatial bin of 2 cm. On the dimension of spatial bins, the place maps were smoothed with a Gaussian kernel with a STD of 5 cm; on the dimension of laps, the place maps were averaged every two laps. Then the processed place maps from different tracks and sessions were concatenated, and projected to a 2D space with UMAP transform. The parameters of the UMAP transform were n_neighbors = 50, min_dist = 0.05, metric = 'cosine'.

## Place map similarity across sessions

Place map similarity was computed only on the first and the last 50 cm segments of linear tracks to match the length of the stationary segments of detoured tracks across all tracks. The middle segments were excluded, as on detoured tracks, there was no explicit correspondence between detour and non-detour sessions. The tuning curves on the first and last segments were concatenated, and the cosine similarities

were computed across sessions. For a given track, direction, and session pair, only the cells with peak rate over 2 Hz and within session stability over 0.8 in at least one session were further considered.

## Cell assembly detection and activation

The procedure for cell assembly detection was adapted from previous studies[32,58]. Briefly, spike activities were time binned (20 ms) and z-scored, followed by the principal component analysis to identify eigenvalues and eigenvectors. If eigenvalues exceeded the theoretical upper bound predicted by the Marchenko-Pastur law[58], those modes were defined as cell assemblies. In the eigenvector of significant cell assemblies, cells with weights over two standard deviations above the mean were defined as significantly contributing cells to that assembly.

In order to study the activation of cell assembly during run in the same session or different sessions, the activity during run was similarly binned at 20 ms for each cell, and subsequently Z-scored. The weights (eigenvectors) of the significant principal components were used to calculate the activation strength of cell assemblies. First, the projection matrix was constructed as the outer product of the weight vector of the cell assembly. The diagonal elements of this projection matrix were set to zero to eliminate the contribution from single neuron activities. This corrected projection matrix was quadratically multiplied by the normalized spike activities to get the activation strength of cell assemblies.

To compare the cell assemblies' similarity across sessions, we first detected significant cell assemblies for each run session. Then we ran a Pearson correlation between cellular patterns (cell's contributions to the cell assemblies) of all the significant cell assemblies across sessions. We ended up with a matrix containing pairwise assembly similarities. We found the maximum value within rows and then averaged row maximum values to get an assembly similarity measure for a session pair.

## Plastic/elastic measure

To evaluate whether the detour impact would reverse or stay after the detour session ended, a plastic/elastic measure was introduced. The cells' or assemblies' tuning curves on the first and the last 50 cm linear segments were computed and concatenated within each pre-detour, detour and post-detour session. The plastic/elastic measure was defined as the cosine similarity between post-detour and detour tuning curves minus the cosine similarity between post-detour and pre-detour tuning curves. The measure has a range from −1 to 1. Given a complete remapping from pre-detour to detour session, the −1 plastic/elastic measure represents that post-detour has the identical tuning curve with pre-detour (i.e., elastic), while +1 plastic/elastic measure represents that post-detour has the identical tuning curve with detour (i.e., plastic). If the tuning curves are stable or random across sessions, the measure will have a value close to 0.

## Multivariate linear regression

A multivariate linear regression model was used for predicting the post-detour tuning curve based on pre-detour and detour activity using the Matlab function mvregress. Tuning curves on the first and the last 50 cm segments of tracks were concatenated and used, as those parts were preserved across sessions.

The post-detour activity for a given cell was modeled with the following equation:

$$\text{Post} = c_0 + c_1 \times \text{Ave} + \beta_1 \times \text{Pre} + \beta_2 \times \text{Det} + \beta_3 \times \text{OT} + \beta_4 \times \text{T1} + \beta_5 \times \text{T3}$$

$$(5)$$

where Ave represents averaged tuning curve across all the cells over pre-detour and detour sessions. This term accounts for the overall features of tuning curves, such as firing rates are higher near track ends. Pre and Det represent that cell's tuning curves on pre-detour or

detour sessions. OT represents the curve on the opposite parallel track during detour session, and T1 T3 represents curves on those tracks during the detour session.

To evaluate how well each regressor predicts the dependent variable, we compared the model residuals between two models. The baseline model is the full model, and the model residual $resi_{full}$ was computed as the Frobenius norm of the residual matrix. Then, for each regressor of interest (Pre, Det, OT, T1, and T3), we removed it from the full model and recomputed the model residual $resi_{\beta_i}$. The variance explained by each regressor $var_{\beta_i}$ was defined as:

$$var_{\beta_i} = \frac{resi_{\beta_i} - resi_{full}}{resi_{full}} \qquad (6)$$

To take regressors' variance difference into consideration, we shuffled the observation identity ($n = 1000$) and recomputed the explained variance. The actual variance explained was compared with the shuffled distribution to get a significant measure.

### Predicting drift and stable sequences in Run4 from Sleep1 or Sleep3

Drift and stable place map sequences in Run4 were extracted from cells with peak firing rate over 2 Hz in Run4, and Run1 vs. Run4 place map correlation smaller than (drift) or larger than (stable) 0.3. Cells were sorted based on their peak firing rate locations in Run4 (directional sensitive), and the probability of run place map sequences were computed from the Markov transition probability matrices estimated from Sleep1 or Sleep3. The run sequence probability was normalized by sequence length (geometric mean) and compared with probabilities from random sequences to get a percentile value.

To predict the theta sequence in Run4, theta cycles with more than 4 active cells on detoured tracks were extracted. In those theta cycles, we separately generated stable or drift theta sequences with stable or drift cells detected with the criteria mentioned above. The probabilities of those sequences were computed from the Markov transition probability matrices estimated from Sleep1 or Sleep3.

### Brain state classification during run session

During running sessions, theta and ripple brain states were identified. The theta ratio was determined by comparing the instantaneous power in the theta range (7–10 Hz) against the power in its adjacent frequency ranges. The theta brain state was defined as the epoch when the running velocity was higher than 10 cm/s and the theta ratio (7–10 Hz)/(2–15 Hz) was greater than the median value of the entire session.

The ripple state was defined based on ripple detection, where the raw LFP was filtered in the 140–220 Hz range. The mean ripple power in this band was computed, and periods exceeding 5 standard deviations above the mean were considered as ripple candidates, then the ripple duration was extended to when the ripple power crossed the 2 standard deviations above the mean threshold. Ripples with a duration shorter than 30 ms or longer than 200 ms were excluded.

### Representing alternative track

Representing the alternative track was investigated by performing Bayesian decoding during a run with 40 ms time bins. Five place maps were concatenated and the decoding probability was distributed among those tracks: (1) the first 50 cm segment on detoured track (left unchanged); (2) current segment (150 cm detour segment during detour run or 50 cm current middle segment during reversal run); (3) the last 50 cm segment on detoured tracks (left unchanged); (4) alternative segment (50 cm pre-detour mobile segment during detour run or 150 cm detour segment during reversal run); (5) control segment on the opposite of the maze with the same length with the alternative segment. To alleviate the impact of firing rate difference on decoding probability, place maps from all the segments were rescaled to have the same mean firing rate. Only time bins with more than 3 active pyramidal cells were used in further analyses.

### Theta phase modulation of detour and middle mobile segment probabilities

During theta states, the influence of theta phase modulation was examined by assigning the decoding probability to the midpoint of the decoding window. The theta phase of that point was determined using the Hilbert transform. We used two methods to investigate the theta phase modulation. We computed the average decoding probability of different segments across the theta phase, or we estimated the distribution of theta phase of strong representation epochs (probability over 0.9) and tested the uniformity of the distribution using the Rayleigh test.

During plotting examples, a 90% overlapping window was used. This overlapping window was also employed in the theta modulation analysis since a 40 ms window was too coarse for theta phase analysis. To address the lack of independence, the degrees of freedom were corrected (10% of the window number) in computing the standard error of the mean during probability versus theta phase plotting. In other analyses (Fig. 6, except for panels b, d, h, and j), a non-overlapping decoding window was used.

### The spatial-temporal pattern of the decoding probability in the alternative segment

The spatial-temporal pattern of the decoding probability in the alternative segment was obtained by averaging the decoding probability based on the lap and linear position on the current track. Principal component analysis was conducted on this 2D matrix of probability (laps × linear position), and the first principal component was plotted.

During the detour run, the spatial pattern of the principal component was analyzed to determine if the probability of the pre-detour mobile segment near the corners where the detour was entered (< 20 cm) was significantly higher compared to positions further away from those corners (> 40 cm). During the reversal run, the spatial pattern of the principal component was examined to determine if the probability of detour on the two unchanged linear segments was significantly higher than that of the middle segment.

### Identifying detour-, mobile-, and reversal-representing epochs during detour and post-detour run

To identify detour-, mobile-, and reversal-representing epochs, decoding probabilities of detour, mobile, or reversal segments were computed by summing over probabilities on corresponding spatial bins. The sum of probabilities was smoothed with a moving average window of 40 ms. The representing epochs were defined as the continuous period with a sum of probability over 0.5 and the animal's running speed larger than 10 cm/s. The cells' firing rates, as well as spike sequences, were extracted from those epochs for further analysis in Fig. 6.

### Statistical analysis

Parametric statistical tests (one-sample and two-sample Student's $t$ tests) were performed on data that did not violate the normality assumption. Otherwise, non-parametric statistical tests (Kruskal-Wallis ANOVA followed by Tukey-Kramer multiple comparison correction, Wilcoxon ranksum test or the Wilcoxon signed rank test) were performed. The significance of counts in data versus a theoretical value was tested by the one-sided Binomial test (data larger than chance level). The significance of differences in counts between two datasets was tested by the Z-test for two proportions. A $P < 0.05$ was considered significant. ***$P < 0.001$, **$P < 0.01$, *$P < 0.05$, n.s. = not significant. Tests were two-sided unless specified otherwise. UMAP analysis was run in Python v3.8, all other analyses used MATLAB (R2017b; MathWorks). In visualization, error bar plots depicted the standard error of the mean. Violin and box plots displayed the data distribution with

whiskers representing the 5th to 95th percentile range; the box represented the 25th to 75th percentiles, and the center dot indicated the median value.

## Reporting summary

Further information on research design is available in the Nature Portfolio Reporting Summary linked to this article.

## Data availability

Data used in this study are available from the corresponding author upon request. The reported data are archived on file servers at the Yale Medical School. Source Data are provided with this paper. No clinical datasets or genetic data have been used in this study. Source data are provided in this paper.

## Code availability

The custom codes specific to this study that are needed to interpret, verify, and extend the research in the article can be accessed at: https://github.com/GDYlab/GDYlabcode. Additional codes will be available upon request from the corresponding author.

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

## Acknowledgements

We thank K. Liu and U. Farooq for help with data analysis and all the Dragoi lab members for helpful discussions. This work was supported by NIH grants R01NS104917, R01MH121372, and R35NS132342 to G.D.

## Author contributions

Y.Z. and G.D. analyzed the data. J.S. and G.D. collected the data. G.D. conceived and designed the study. G.D. and Y.Z. wrote the manuscript.

## Competing interests

The authors declare no competing interests.
