## [Transparent Peer Review file · Nature Communications]

Generative emergence of non-local representations in the hippocampus

Corresponding Author: Dr George Dragoi

Version 0:

Reviewer comments:

Reviewer #1

(Remarks to the Author)

In this study, Zhou Sibille and Dragoi, record from rats during sleep and running on a square maze with changing segments. These changing segments are labeled as “detours.” The authors perform many different analyses on this dataset. The manuscript has three main findings: 1) While the detour causes partial remapping of the place cells, the sequential firing of neurons on the detour segments is predicted by the preceding sleep epoch. 2) The detour segment place fields overlap with the post-detour place field and these continue to be replayed after the detour has been removed. 3) There are phases of theta during which a Bayesian decoder flickers to track segments that are not currently present. None of these findings are in themselves extremely new, since preplay, remapping, and remote replay have been shown before in different kinds of experiments. However, it does have some new insights, and due to the novelty of the overall experimental design, this study could add meaningfully to existing knowledge if the manuscript is revised to highlight the key observations and new insights in a more straightforward way and account for potential confounds.

Main comments:

This manuscript proved to very challenging to review, as there are many complex analyses performed for which the motivation is not always very clear. Some of the analyses seem to largely confirm previous findings. The data is illustrated in only highly processed forms, which make it very difficult to relate to some of the existing literature. I will try to go through these in the order in which they appear in the manuscript, rather than order of importance.

I had a very hard time to understand Figure 2. The authors claim that theta sequences are present on the first lap, but I find this hard to see from the plots. Some theta sequences are evident during the run shown in Fig 6b. Why not also show the first few laps in the same manner here in this figure? Another way theta sequences have been shown in past papers is through the cross-correlation spike histograms. Why not provide some of these here?

I also struggle to understand the authors claim that these theta sequences are pre-configured. First, previous studies have examined this question and found contradicting observations. Skaggs and McNaughton (Hippocampus 1996) who first showed theta sequences in the cross-correlation histograms showed that these same sequences are present in sleep after but not in sleep before (Science 1996). I think this same observation was confirmed some years later by Wikenheiser and Redish (Hippocampus 2013) who came to the same conclusions. What did those studies miss? The cross correlation histograms and temporal bias analyses used in those studies are very straightforward and easy to understand. Please use them.

Second, if the theta-sequences are pre-configured, then why are they not observed during the preplays on the track? Do you have to go back further in time? If that is the case, how far back in time is needed to find these preplays? Preplays preceding Lap1 could be added to Fig 2A to illustrate that.

Some of the results in the later figures seem to be in contradiction with those in the earlier figures, perhaps because there are so many different analyses. For example, Fig 5D seems to indicate a non-negligible similarity in the place fields before

and after the detour. Some shuffles should be added here to see how does this relate to Fig 1D?

Also Fig 5E results indicate that pre-detour curves and detour curves overlap after the detour has been removed. A similar analysis to this could also be performed to test how much of the detour tuning curves are explained by pre-detour tuning curves. In one of the sample cells shown in Fig 5D there is overlap between the tunings on T2 and T2Det, and it looks like from Fig 1F that a lot of the same cells are active in both. This is also worth looking at: how many cells have fields both before and during detour, even if they seem remapped, and how does this percentage compare with other locations? And also, what is the correlation between firing rates of cells between Pre-detour and Detour (like in the Fig 7H scatterplot)?

In Fig 6 and 7, the analyses is focused on the observation of flickers in the Bayesian posteriors obtained at different time points. This is potentially interesting, but it could also arise due to noise or lower firing rates at the peak of theta. Decoding errors are inevitable, especially for cells that have multiple and overlapping place fields. These decoding errors are probably more likely at the peak of theta, where there may be fewer cells firing. Why should these decoding errors be taken seriously? Is it not expected that the overlap between Det and Mob place representations will lead to occasional flickers in the decoder when the animal is on the Detour segment? Or flickers to Det when the animal is on Ret segments due to the remapping shown in Fig 5E?

It could also be examined when there are decoding errors to the control segment or other locations. Even if these don't happen as much, because there is less overlap in the place fields, they probably still happen at the peak of theta.

Finally, I don't think that Markov chain is appropriate for analyzing these firings. The transition matrices that are shown have strong vertical stripes in them. This means that the same cell is very likely to be the "next one" regardless of what else is happening. This is because some cells are higher firing than others, so most of the analysis is influenced by which cells are higher firing rather than which order they fire in. If there is really a predictable firing order, then this could be shown with the cross-correlation spike histograms, discussed in the previous comment.

Minor comments:

The use of the term "plasticity" is a bit confusing. Usually plasticity refers to synapses but that is not obvious to be the case here. Is there direct evidence that increased replay versus preplay results from synaptic plasticity, and not maybe short-term changes in membrane excitability?

There seem to be some grammatical issues where "theta sequence" (e.g. L 164), state (e.g. L350), and oscillation (L360) are used in the singular without a "the" in front.

The need for Figure 2C-E were not clear to me. It seems these analyses are largely to confirm previous findings about phase-precession versus theta sequences that are not in dispute. If so, perhaps these should be moved to the supplement.

I also didn't understand the motivation for Figures 5F-H. These could also not be put in a supplement?

The terms used to describe the different maze segments keep changing throughout the manuscript. E.g. T2L is later referred to as the Reversal Segment. Run1 is sometimes referred to as the pre-detour segment, and Run4 is sometimes called the reversal run. If this can be simplified, it would make the study easier to follow.

Labels in Fig 1D are ambiguous. What is T13? Does it mean either T1 or T3? If so why is this labelled differently in Fig 1C. What is OT in Fig 1E?

What is the running direction in Fig 1F? Or is it combined across both directions?

The arrow is not mentioned in Figure 4a legends.

(Remarks on code availability)

There wasn't code related to the paper and I could not install and run it.

Reviewer #2

(Remarks to the Author)

This manuscript by Zhou et al describes their study on how novel sequences within individual theta cycles (theta sequences) emerge during a novel detour experience and how they interact with sequences during awake and sleep replay (p/replay sequences) to impact place cell maps. The authors conclude that the rapid emergence of theta sequences "critically" depends on "internal dynamics" such as preplay and the flickering of both novel and familiar sequences during navigation reflects the plasticity and stability (termed "elasticity") of place maps in the hippocampal CA1. The study of p/replay in the dynamics of theta sequences and place cell maps during navigation is important. The experiments are well done and data analysis is comprehensive. However, there are several major concerns on the conclusion regarding the role of preplay, which is difficult to draw from the data currently presented.

1) The authors conclude that (detour) theta sequences appeared earlier than replay (Fig.2a,b). Whereas this is not surprising based on the literature, it is unclear what this comparison means, given very different definition and the different criteria for

detection. Despite the authors' lengthy discussion on the equalization of false positives in their detection, it remains unconvincing because it is unknown whether same percentage of false positives should be expected from random data sets.

2) The authors conclude that theta sequences were "decorrelated" from phase precession (Fig.2d,e), because disrupting precession by jittering sometimes produced significant theta sequences. However, the raw data presented clearly show a drop in the quality of theta sequences (Fig.2d). Second, the grouped result shows that large jitters (>70ms) disrupted both precession and theta sequences (Fig.2e), consistent with a tight correlation between precession and theta sequences. The authors' conclusion seems to rely on the statement that sometimes theta sequence was "significant ($p < 0.05$)" when precession was "not significant". This inference is not easy to understand. The p-values were determined differently and not necessarily linearly scaled with jittering sizes. In addition, conceptually phase precession by definition leads to theta sequences (Fig.2d top). Therefore, although there may be a contribution from non-phase-precession factors, the data presented seem more supportive for the opposite: phase precession is a major contributor to theta sequences.

3) The authors conclude that theta sequences were predicted from the preplay during sleep by a Markov process (Fig.3). However, the argument seems circular. The transition matrix (Fig.3f) was constructed from the "forward preplay" already determined. These highly selected data points were only a small proportion of activities during (pre) sleep (Fig.3c), and by definition would "predict" theta sequences, because forward preplay and theta sequences had same firing orders among cells.

4) The conclusion in the abstract saying the internal dynamics "critically support" rapid sequence coding. The phrase is too strong for the data presented. It is unknown whether preplay was causal or even contributed equally as novel experience.

5) In many of the plots, it is unclear what each dot (sample) is.

6) Fig.3e, were there 57 cells? But the line thickness is not consistent with such a big number.

(Remarks on code availability)

Version 1:

Reviewer comments:

Reviewer #1

(Remarks to the Author)

The authors provided a revision that rebuttal that nicely answered some of my concerns. I thought Fig 2h was a nice addition, and the removal of some of the panels makes the main figures more straightforward. And I appreciate they clarified the segments and their overlaps in Fig 5 analyses. The controls they did for the Bayesian analysis in Fig 6 appear quite reasonable. They also provided a good explanation that the Markov method does capture something other than just firing rates since it only works on some parts of the maze. However, for me some critical issues remain.

First, with regards to why the CCG doesn't work in this task, their response makes sense: the rat runs in two direction which cancel out any directionality in the firing order and there are many maze sections that have different patterns. "method that averages across the whole sleep reflecting multiple experiences, like CCG, fails to capture the one experience-related sequential structures that are only expressed in a part of sleep frames (6-7%)". However, I had not realized that so many of their arguments regarding "prediction" relied on this subselection of "forward detour preplays". This really supports the second reviewer's major concern about the circularity of the related claims. If 6 percent of events are like the theta sequences but another 6 percent are the exact reverse running direction, then I don't think anything can be said to have been predicted from these periods combined. I read carefully the authors rebuttal on this point, but unfortunately I did not find those arguments persuasive. E.g. I don't understand the authors assertion that preselection of forward preplays makes "no assumption of their sequentiality". If the overall activity does not reflect any particular temporal pattern, then to me it doesn't make sense to suggest there are existing network biases that are predictive of the later theta sequences. I think it could be fine to detect what proportion of events in sleep, etc, qualify as preplay and report the proportions, as the authors do in many points, and to report these patterns as being *present* before the detour segment. However, the claim that these patterns were *predictive* is indeed circular, since they could not be detected without using the specific place-field sequence on the detour.

In the rebuttal of this point, the authors also refer to some tuplet analysis from an earlier paper. I read this paper and really tried but it was very difficult to understand. I still don't really understand what is a 2-neuron tuplet and how it differs from the pairwise CCG peak. If two cells reliably fire in some order, that should be very clear in the CCG. What does it the tuplet mean then? Maybe it's picking up misclustered bursts or something other than reliable patterns?

The rebuttal also provide some new rank-order correlations between pre-sleep and theta. This analysis is promising but seems preliminary and should be better illustrated. It seems concerning that the rank-order correlation does not do very well in identifying the Bayesian decoded preplays and replays, since Bayesian decoding is at the heart of this paper. It also left unclear how both the forward and reverse detour directions could be matched to these preplays.

Another point I thought Reviewer 2 had an excellent comment about the the relationship between phase precession and theta sequences. To me it sounds like the jitter mainly tells you about the temporal sensitivity of the measures used. So if phase-precession measures used are very sensitive to jitter (maybe because they pool all trials instead of using single trials—that has been found that single trials are more robust than averages) and also perhaps because of sawtooth shape of theta waves impacting phase measurements with the Hilbert transform (Belluscio et al), then it could be that small amounts of jitter ruins phase precession but not theta sequences. But that doesn't mean that the two phenomena are unrelated (even if that kind of argument was used in some papers before). One is likely just based on measures that more sensitive to jitter than the other.

Around L379, the study concludes “the novel detour differentially impacted the associated T2/T4 tracks stronger compared to the non-detoured T1/T3 tracks suggesting its experience was different than a simple exposure to a novel track environment.” However, the study reports that there were actually more elastic cells, defined as those with place-fields that return, on T2/T4 than on T1/T3. (Note also that the claim on L382 about the firing rates is only true for T2/T4, not T1/T3). This suggests a rather surprising lasting effect of the detour on the other tracks. So either the plastic/elastic measures are problematic or this conclusion is over simplified.

One other point of confusion for me. The study notes 5 rats with 2 detour sessions per rat, summing to 10 detour sessions. And yet there are often 16 or more data points in the figures. What am I missing? Maybe it would help if the figure legends made note of the n's used in the various panels.

Finally, the manuscript makes a distinction between its findings and those reported by Feng, Silva, and Foster J Neurosci 2015. I looked back at this paper, and I noticed that they emphasized the rapid improvement in the theta sequences after the first lap, but did not report on whether or not the first lap itself was also significantly ordered. Indeed the first lap quadrants shown in that paper are not much different from those in this one. This could be an important detail in reconciling the two studies.

(Remarks on code availability)

I just glanced through the code quickly. Seems to be appropriately detailed.

Reviewer #2

(Remarks to the Author)

This revised manuscript has mostly addressed my questions. Although I feel like some of the responses are not super clear, I think the revision is adequate.

(Remarks on code availability)

Version 2:

Reviewer comments:

Reviewer #1

(Remarks to the Author)

I don't agree with everything in the rebuttal letter, but I think the authors have done a good job clarifying their findings and methodology and the manuscript is clear enough for other readers to reach their own conclusions. Overall, I think this is a great paper, so I give congratulations to the authors.

(Remarks on code availability)

Response to reviewers' comments for manuscript NCOMMS-25-06854-T

Reviewer #1 (Remarks to the Author):

In this study, Zhou Sibille and Dragoi, record from rats during sleep and running on a square maze with changing segments. These changing segments are labeled as "detours." The authors perform many different analyses on this dataset. The manuscript has three main findings: 1) While the detour causes partial remapping of the place cells, the sequential firing of neurons on the detour segments is predicted by the preceding sleep epoch. 2) The detour segment place fields overlap with the post-detour place field and these continue to be replayed after the detour has been removed. 3) There are phases of theta during which a Bayesian decoder flickers to track segments that are not currently present. None of these findings are in themselves extremely new, since preplay, remapping, and remote replay have been shown before in different kinds of experiments. However, it does have some new insights, and due to the novelty of the overall experimental design, this study could add meaningfully to existing knowledge if the manuscript is revised to highlight the key observations and new insights in a more straightforward way and account for potential confounds.

We thank our reviewer for finding our study "does have some new insights, and due to the novelty of the overall experimental design, [...] could add meaningfully to existing knowledge". We are especially grateful for the reviewer's constructive suggestions in the main comments.

Main comments:

This manuscript proved to very challenging to review, as there are many complex analyses performed for which the motivation is not always very clear. Some of the analyses seem to largely confirm previous findings. The data is illustrated in only highly processed forms, which make it very difficult to relate to some of the existing literature. I will try to go through these in the order in which they appear in the manuscript, rather than order of importance.

We appreciate our reviewer's patience and effort in helping us improve our manuscript. We have reorganized our manuscript and rearranged figures based on reviewer's suggestions to improve readability and clarity. In general, we employed established approaches widely used in the field, such as place map correlation, population vector correlation, Bayesian decoding, and Markov chain (please see our response corresponding to reviewer's critique to Markov chain below) applied to our new design. The novelty and complexity of our task structure was largely reflected in our explaining how those methods were applied to our analysis, which is now revised to better justify all the analyses. To further improve clarity, we have included illustrative cartoons that better motivate and summarize our results.

I had a very hard time to understand Figure 2. The authors claim that theta sequences are present on the first lap, but I find this hard to see from the plots. Some theta sequences are evident during the run shown in Fig 6b. Why not also show the first few laps in the same manner here in this figure? Another way theta sequences have been shown in past papers is through the cross-correlation spike histograms. Why not provide some of these here?

Based on reviewer's constructive suggestion, we have added decoding results across individual theta cycles to illustrate early theta sequences (laps 1-2). We also included distribution of quadrant ratios across the laps, as well as pairwise temporal bias analysis, where the bias was derived from neuronal pairwise spike cross-correlogram (CCG), and we correlated the bias across different temporal scales (i.e., behavioral vs. time-compressed scales), or correlated it with the distance between place cells bias, as in Dragoi and Buzsaki, Neuron (2006).

We agree with our reviewer that the spike CCG based analysis has been one illustrative way to study sequential structures. Related to the reviewer's next comments on CCG as used in Skaggs et al., Hippocampus (1996), we now discuss why we didn't use it in our initial submission and why we think it

might not be appropriate to use under certain circumstances. The spike CCG was used in Skaggs et al., Hippocampus (1996) paper where they reported on single cell phase precession and looked at pairwise sequential orders based on spike CCG. They used examples of a total of 4 cells showing temporal bias at theta and behavioral timescale and a summary cartoon figure where “each part was drawn by hand”, but no analysis or proof showing sequential activation of place cells at compressed theta timescale matching the place field sequence. Specifically, in that paper they (also nobody else prior) did not conduct a direct correlation analysis to show that the degrees of bias across the 2 timescales are correlated, nor a correlation between the time-compressed temporal bias and the corresponding place fields bias. These 2 critical analyses were performed for the first time in Dragoi and Buzsaki, Neuron (2006), which demonstrated the existence of hippocampal theta sequences, in CA1 as well as in CA3. Without the time-compressed bias vs. corresponding place field bias correlation (equivalent to a sequence analysis), pairwise spike ordering alone is insufficient to reconstruct theta sequences that would align with place field sequences.

The correlation between the bias of place fields and bias of corresponding theta-scale CCG was studied in depth by Dragoi and Buzsaki, Neuron 2006. As our reviewer pointed out, the correlation of CCG bias between sleep and run (both calculated in the temporal domain) was used by Skaggs and McNaughton, Science (1996) and Wikenheiser and Redish (Hippocampus, 2013) aiming to study sequences during theta rhythm and sleep. While we agree the correlation of CCG conveys information that is related to sequential structure, they also have key limitations and were subject to several critiques, which we discuss below:

1. In a direct critique of Skaggs and McNaughton, Science (1996) method, George P. Moore et al., Science (1996) pointed out that their method was actually a direct measure of temporal bias and not of a sequence, a fact which was acknowledged by Skaggs & McNaughton in their response to the critique, Science (1996). The difference between temporal bias and sequence can arise due to the complex spike pattern of a single cell. When we try to assign each cell one representative spike time, we inevitably lead to inaccuracy and disagreement of what is the actual sequence, due to the other spikes the cells emitted within the epoch of investigation (see Moore et al., Science 1996).
2. Quirk and Wilson, Journal of Neuroscience Methods (1999), tried to reproduce the results from Skaggs and McNaughton, Science (1996) to study the relationship between theta cycle and post-run sleep. They only found positive CCG correlation between cell pairs with high vs. low spike amplitude, which they stated could primarily arise from the misclustering of single-cell bursts as separate cells. Later Wilson’s laboratory developed a template matching method based on ensemble rank order correlation rather than pairwise analysis to study sequences across brain states. In Lee and Wilson, Neuron (2002), where they used a form of rank order correlation to study correlated sequences between place cell order in run and corresponding firing sequences in NREM sleep, they also included a control to take into account the potentially increased CCG correlation in sleep, and concluded that their sequence correlation cannot be explained by this pair-wise relation. Their rank order correlation method was also impacted by the issue mentioned in point #1 where one needs to assign a representative spike time for each neuron. However, one important advantage is that this method captures the rich variability of spike dynamical patterns during sleep, while the pairwise CCG method is an average measure which reduces the information to a single value per cell pair for the entire sleep period. We will further elaborate on this point based on our data in our response to the reviewer’s next comment.
3. To resolve the issue mentioned in point #1 and to transition from pairwise to full ensemble analysis, the Bayesian decoding method became widely used in studying sequential structures. It has the advantage of using all the spikes from all the units to decode their representation, and can be

potentially less sensitive to clustering quality (by using cluster-less decoding as in Kloosterman & Wilson, JNeurophys 2014). The decoded run trajectory is also illustrative of what the spikes are representing. Bayesian decoding also has several disadvantages: it assumes cells are independent and have Poisson-like spikes (not true); it requires using a time bin, thus, has limited temporal resolution; to determine whether the decoding posterior probability is significantly sequential, we need to use shuffle methods that do not always match our expectation and can introduce false positives or negatives.

Considering all those factors, now we integrate multiple analytical approaches to study the sequential structure in early detour laps and sleep. By combining these complementary methods, we aim to provide a more comprehensive and convincing interpretation of our results.

I also struggle to understand the authors claim that these theta sequences are pre-configured. First, previous studies have examined this question and found contradicting observations. Skaggs and McNaughton (Hippocampus 1996) who first showed theta sequences in the cross-correlation histograms showed that these same sequences are present in sleep after but not in sleep before (Science 1996). I think this same observation was confirmed some years later by Wikenheiser and Redish (Hippocampus 2013) who came to the same conclusions. What did those studies miss? The cross correlation histograms and temporal bias analyses used in those studies are very straightforward and easy to understand. Please use them.

We understand our reviewer's concern in interpreting that our results appear not consistent with the previous literature. We believe this discrepancy highlights an important issue: the potential limitations of CCG methods and their sensitivity to experimental design (please also see our response to the previous comment above). In our data, we computed the cell-pair CCGs during sleep and found they were not correlated with CCGs during run theta cycles either during pre-run sleep or post-run sleep. As we mentioned above, there are critical dissociations between the CCG-based and spike sequence analyses, as first noted by the previous literature (George P. Moore et al. Science, 1996; Quirk & Wilson, Journal of Neuroscience Methods, 1999; Lee & Wilson, Neuron, 2002). Apart from that, we think the main reasons that could cause the difference between our current results and the previous studies are: 1. difference in experimental design; 2. CCG analysis is a data-averaging approach, which ignores the variability of spike patterns during sleep. We will elaborate on these reasons below. First, we want to point out some unique features in those two studies.

1. In both studies (the McNaughton and Redish groups), their animals ran unidirectionally on one novel maze with continuous, unsegmented structure. This means their animals only had a sequential experience with only one direction during the run. As we will explain below, this influences how effectively CCGs can capture sequence-related neuronal activity.
2. In both those studies, their animals experienced shorter pre-run sleep compared with post-run sleep. First, the limited spiking in the short sleep itself can compromise the overall quality of CCG analysis due to reduced sampling, primarily in pre-run sleep. Second, running experience increases the overall firing rates of cells from pre- to post-experience, which will further favor pattern detection in post-run sleep compared with pre-run sleep (also shown in Battaglia et al., Neural Network 2005). In Skaggs & McNaughton (Science 1996), the pre-run sleep was 30 min, and the post-run sleep was 30-60 min. During analysis, they picked a 15 min sleep epoch to run the CCG analysis. The overall firing rate is lower in sleep compared with running, and the 15 min sleep may not contain adequate multi-unit activities (MUAs) to produce reliable CCGs, particularly in pre-run sleep when rates are lower than in post-run. This is evident in their Fig. 1B, where the pre-run CCGs appear sparse and may not reflect multi-unit activity. In Wikenheiser and Redish (Hippocampus 2013), the pre-run sleep was 10-15 min and post-run sleep was 20-30 min. The short epochs of sleep could be insufficient to collect enough spikes for a robust CCG analysis, especially for the shorter pre-run sleep while the twice longer post-

run sleep epoch together with the significant post-run increase in firing rate compared to pre-run might sample enough neuronal spikes to compute reliable CCGs.

Next, we discuss why the CCG analysis would not work well in representing multiple experiences (e.g., in our experiment). We hope the reviewer agrees with us that hippocampal activities during sleep are highly variable and that exposure to multiple experiences overall increases this variability. One recent study from our group (Farooq and Dragoi, 2024 Nature Communications) showed that activity in sleep frames formed several ensemble clusters, called repertoire, and that this repertoire was reduced in rats deprived of early-life experience of Euclidian geometry features. In another study (Liu, Sibille, Dragoi NatNeurosci 2024) we showed this repertoire can separately represent at least 15 tracks x 2 directions via sets of p/replay sequences in pre- and post-experience sleep. Unlike head direction or grid cell systems, where the neural dynamics are found to have a consistent low dimensional geometric structure across run and sleep (e.g., torus manifold, Chaudhuri et.al. 2019, Gardner et.al. 2022), hippocampal dynamics during sleep were never found to have a simplified geometric structure consistent with run, to the best of our knowledge. Consequently, a method that averages across the whole sleep reflecting multiple experiences, like CCG, fails to capture the one experience-related sequential structures that are only expressed in a part of sleep frames (6-7%).

The CCG correlation analysis could work better if there were a single novel experience that significantly impacted the circuit, and the potential consolidation process would dominate the post-run sleep. That could bias the average measure in post-run sleep toward correlating with that experience. On the other hand, there would be no strong reason for the pre-run sleep to be dominated by one given future experience. That might explain the Skaggs and McNaughton Science (1996) and Wikenheiser and Redish Hippocampus (2013) findings, where there was one direction taken on a single maze novel experience. However, in our experiment, this was not easy to observe, because rats ran in two directions during two detour experiences (i.e., a total of four novel run sequences) before/after any post/pre-detour sleep. Moreover, we found the CCG patterns between the first and second detour were uncorrelated even for the same direction during run, and the CCG patterns across two directions in the same detour were negatively correlated. More likely, those run representations were distributed across sleep frames and the averaged result won't particularly match any of those.

To apply the CCG analysis, we selected frames in pre- or post-run sleep that were significant as forward preplay or replay for a given direction and detour experience, and computed CCG from those selected frames. After selecting significant frames, we found a significant positive correlation between theta-scale CCG bias during run and CCG bias in pre/post-run sleep. The different CCG patterns based on our selected frames also support our statement of strong neural variability in sleep. These results were summarized in Extended Data Fig. 7.

In this sense, we consider that averaging over the whole sleep for CCG correlation analysis is a method with relatively high detection threshold and extremely low representation and discrimination capacity (i.e., not able to support detection of multiple distinct representations).

Second, if the theta-sequences are pre-configured, then why are they not observed during the preplays on the track? Do you have to go back further in time? If that is the case, how far back in time is needed to find these preplays? Preplays preceding Lap1 could be added to Fig 2A to illustrate that.

We understand the potential confusion and want to clarify that we are not stating that detour replay must emerge later than theta sequence. Rather, we are simply stating that in our experiment, waking state replay is observed primarily after the emergence of theta sequences during the first 1-2 laps of novel detour. As our reviewer pointed out, given there is preconfiguration of theta sequence, one could observe it at any time, including during preplay on the detour track, especially if we would have

enforced a resting period before the first detour lap in the maze (which we did not). We observed several detour preplay events during waking rest state, especially during pre-detour sessions on different tracks during adequate waking rest frames. The overall number of waking rest preplay of detour while on other tracks during pre-detour sessions was significantly higher than chance level (Fig. 3c). During the detour session, based on our animals' spontaneous behavior, the resting epochs on the detour track prior to the first lap were very limited. Within those frames, we can still detect individual detour preplays, while the comparison against shuffle was not significant due to the very limited samples. We have added those infrequent examples and their analysis to the Extended Data Fig. 6. However, we focused our analysis on the sleep preplays occurring in the immediately pre-detour sleep session in a sleep box located outside the maze and prior to experience of detour. Also to have a better understanding of the temporal dynamic of preconfiguration and potentially related to our reviewer's question "[...] *how far back in time is needed to find these preplays*", we investigated the distribution of detour preplay across sleep and waking rest brain state before the first detour experience. We found a relative stable distribution of preplay over time (Fig. 2h).

Our main motivation in this study was to explain the rapid development of compressed theta sequence. We wanted to investigate what could drive its development, and hypothesized that experience-induced waking rest replay could be one candidate mechanism. Because we wanted to study the potential contribution of experience-induced waking rest replay, in this analysis we only included waking rest frames after the animal has stepped on the detour segment. A convincing argument for stating that theta sequence was not driven by experience-induced replay relied on the fact that replay emerged later than theta sequence in the novel detour. We showed that in our experimental design and given animals' spontaneous behavior of not resting a lot during early laps, experience-induced replay emerged statistically later than theta sequence. In a minority of cases, replay developed earlier than theta sequence, but at the single event level it was possible that those replays were mainly expressions of pre-configuration rather than induced by the detour experience, particularly given they were already observed in the pre-detour sleep. In the revised manuscript, we modified the text to make our logic and motivation clearer.

Some of the results in the later figures seem to be in contradiction with those in the earlier figures, perhaps because there are so many different analyses. For example, Fig 5D seems to indicate a non-negligible similarity in the place fields before and after the detour. Some shuffles should be added here to see how does this relate to Fig 1D?

We thank our reviewer for raising this point. We assumed that by "indicate a non-negligible similarity in the place fields before and after the detour" the reviewer meant to ask "[...] before and during the detour" (blue line in the original Fig. 5d, now revised Fig. 4d), since Fig. 1d only refers to pre-detour vs. detour analyses. If the reviewer indeed referred to before vs. after the detour, then we think it's not a contradiction because we compared the neural activity on the same track before and after the detour (without a direct detour impact), and thus the representations should be more similar. We also added the comparison with the shuffle data as requested by the reviewer for the old Fig. 5d (now revised Fig. 4d).

If, however, the reviewer referred to the comparison before vs. during the detour in Fig. 5d (now revised Fig. 4d), please see the following clarification. There appears to be a misunderstanding here because of our initial lack of clarity, also related to several following comments from our reviewer. We have modified the figure to increase clarity, and we added further clarifications in the text. Briefly, the analyses of Fig. 1 and Fig. 5 (now revised Fig. 4) were conducted on different track segments, and we will explain why we did that. In Fig. 1, our motivation was to test whether there was a fully novel representation on the detour segment, or, alternatively, the animal somehow partially used the previous

pre-detour mobile-segment representation (called here and in the revised text 'structured remapping'). We showed that there was complete remapping between the pre-detour mobile segment (50 cm segment in the center of the track) and the detour segment (150 cm U-shape detour segment) measured at single-cell place map and neuronal population vector levels. As suggested by our reviewer in the next comments, in the revised manuscript we also added the single cell firing rate and place fields analysis to describe the same result (Fig. 1e; Extended Data Fig. 1). However, we want to emphasize that during the detour, tracks 2 and 4 (T2/T4) contained more than the U shape 150 cm detour segments. At the start and the end of T2/T4 tracks, the two 50 cm stationary segments were not changed from pre-detour to detour, and we showed that place map correlations on those segments were significantly higher than shuffle (column 1 and column 6 in Fig. 1d).

In the original Fig. 5 (now revised Fig. 4), our motivation was to investigate whether an in-between experience (i.e., the detour) could impact future representations during the post-detour run. When interpreted as a form of representation drift (measured from Run1 to Run4), could the in-between detour experience partially explain the drift? As there are other factors that could contribute to the drift, we needed to compare this effect against control tracks. In our case, we used representation on non-detoured tracks (T1/T3) as a control. We hypothesized that the detour experience would have a larger impact on the detoured tracks (T2/T4) than the non-detour control tracks (T1/T3). At the single cell level, we studied the place map evolution with this detour experience. There was a technical constraint due to matching the 150 cm U-shape detour segment with the pre-detour or post-detour shorter 50 cm middle segment (we have shown in Fig. 1 there is no evidence for structured remapping; now detailed in the revised manuscript). To measure the correlation between tuning curves, the place map vectors must have the same length, so we could only use place maps on the two stationary 50 cm segments on T2/T4, with equal length across the run sessions. The place map correlation is dependent on the vector length; thus, to make the T2/T4 vs. T1/T3 comparison fair, we only used the start and end 50 cm segments on T1/T3 as well. In Fig. 1d we showed for T2/T4 that place maps were significantly correlated (higher than shuffle) on the two stationary linear segments (column 1 and column 6), and in Fig. 5 (now revised Fig. 4) we showed they were less stable compared with T1/T3, and experienced a larger drift due to the detour experience.

Beyond single cell level, pair-wise CCG analysis and cell-assembly analysis did not suffer from this technical constraint and could be applied directly to the entire track. We included these analyses in Extended Data Fig. 13 to show how pairwise spike order and assembly organization were greatly impacted by the novel detour experience, and were further expressed as plasticity from the detour experience in the post-detour sessions. In Fig. 5 (now revised Fig. 4) we still used the single cell place map analysis as we think showing single cell spatial tuning curves can be more illustrative to readers. We added a cartoon to the revised Fig. 4c to emphasize that the measure was conducted on two stationary linear segments and explained why we did that in the caption.

Also Fig 5E results indicate that pre-detour curves and detour curves overlap after the detour has been removed. A similar analysis to this could also be performed to test how much of the detour tuning curves are explained by pre-detour tuning curves. In one of the sample cells shown in Fig 5D there is overlap between the tunings on T2 and T2Det, and it looks like from Fig 1F that a lot of the same cells are active in both. This is also worth looking at: how many cells have fields both before and during detour, even if they seem remapped, and how does this percentage compare with other locations? And also, what is the correlation between firing rates of cells between Pre-detour and Detour (like in the Fig 7H scatterplot)?

Thank you for suggesting this analysis. The relationship between firing rates and place fields in the pre-detour and detour segments is important for understanding how cells are recruited to represent the novel environment. We have added the single cell firing rates/place fields analysis between pre-detour

and detour in the revised manuscript. In Extended Data Fig. 1a-b, we looked at how many cells had place fields on both pre-detour mobile segments and detour segments, compared with other middle segment in pre-detour session. In Fig. 1e and Extended Data Fig. 1c we investigate the firing rates correlation between pre-detour mobile and detour segments where in Fig. 1e we used the firing rate percentile and in Extended Data Fig. 1c we used the raw mean firing rates. Overall speaking, the firing rates/activities were correlated between detour segment and pre-detour mobile segment, but not stronger than other middle segments.

In Fig 6 and 7, the analyses is focused on the observation of flickers in the Bayesian posteriors obtained at different time points. This is potentially interesting, but it could also arise due to noise or lower firing rates at the peak of theta. Decoding errors are inevitable, especially for cells that have multiple and overlapping place fields. These decoding errors are probably more likely at the peak of theta, where there may be fewer cells firing. Why should these decoding errors be taken seriously? Is it not expected that the overlap between Det and Mob place representations will lead to occasional flickers in the decoder when the animal is on the Detour segment? Or flickers to Det when the animal is on Ret segments due to the remapping shown in Fig 5E?

It could also be examined when there are decoding errors to the control segment or other locations. Even if these don't happen as much, because there is less overlap in the place fields, they probably still happen at the peak of theta.

The reviewer is correct in pointing out that the Bayesian decoding is not perfect and can induce noise. We believe biologically it is meaningful to explain events/activities even if they have low occurrence or deviate strongly from the mean activity. Here, we include more arguments and analyses to convince our reviewer and the readers that our findings were not merely due to technical noise:

1. First, we need to clarify that in Fig. 5 (now revised Fig. 4) the place map overlap was computed only on two stationary 50 cm segments, while the flickering was defined as representing the 150 cm detour segments or the 50 cm pre-detour mobile segments, that do not correlate/overlap. Overlap was evaluated in Fig. 1, and we don't see stronger overlap between flickered representation and current representation during the detour run.
2. Another important point is that from a technical perspective, we can think of flickering or non-local decoding probability as arising from the overlapping place map that we used in our Bayesian decoder. However, biologically, we also asked what led to this overlapping place map or increased similarity of population activities. The flickering representation itself can be the cause instead of the technically-driven effect. In fact, if animals consistently reactivate a detour assembly at a fixed location on the post-detour run, that will generate stable place fields on the post-detour track, which will appear to encode the current context. Under that circumstance, we would be unable to know what exactly the animal/network was 'thinking' about, apart from those cells consistently firing at that specific location. In our case, as the flicker representation is either weakly or not anchored to the current location, technically we were able to detect and separate them from representations of the current, local context.
3. Based on the previous argument, what we can detect are deviations from the average current tuning, and as the reviewer pointed out, we need to make sure there is something meaningful beyond detection or decoding noise. In our original manuscript, we only included time bins with at least 3 active cells. We also compared the decoded probability versus control segments, and tested whether this phenomenon existed in the pre-detour session. Apart from that, we also identified the spatial-temporal pattern of this flickering representation. To further support our conclusion, we added several new analyses presented in Extended Data Fig. 16:
 - 3.1 We identified time bins with at least 3 active cells and decoding probability of the alternative non-local context (pre-mobile during detour run or detour during post-detour run) higher than 0.9. In that case, we have stronger confidence that that epoch is highly likely to represent the currently

absent but contextually related environment. We found the instance of this flickering representation was significantly higher than other control segments during the detour run and post-detour run, but that was not true during the pre-detour run.

3.2 In terms of theta phase modulation, we computed the theta phase of the time bins (based on time bin center) when the decoding probability was higher than 0.9. For the current context, those strong representation epochs occurred more at theta troughs, while for the alternative context, the strong representation epochs occurred closer to theta peaks. For the control segments, the strong representation epochs had a more uniform distribution, which didn't pass the Rayleigh test of phase preference.

Finally, I don't think that Markov chain is appropriate for analyzing these firings. The transition matrices that are shown have strong vertical stripes in them. This means that the same cell is very likely to be the "next one" regardless of what else is happening. This is because some cells are higher firing than others, so most of the analysis is influenced by which cells are higher firing rather than which order they fire in. If there is really a predictable firing order, then this could be shown with the cross-correlation spike histograms, discussed in the previous comment.

We agree with our reviewer 's observation on the appearance of vertical stripes in the transition matrix and the statement that the results are influenced by firing rates. We have the following arguments and analyses to explain why we still used that, and how we support our conclusion with additional analyses:

1. Our group first used the Markov chain model to predict future sequences in Liu, Sibille, Dragoi, Neuron 2018. There too, the transition matrix had the appearance of stripes. Despite that, we provided clear evidence for both neuronal identity and neuronal order prediction (based on order only shuffle, which preserves neuronal identity) using the model, with the latter predicting pure order of neuronal activation of sequences of around 30 neurons. However, when the length of predicted sequence is very short, like in the case of theta sequences, which are limited to the neurons active within a theta cycle (not along the entire track as in the 2018 study), the Markov chain model indeed predicts primarily which neurons will be activated together and less so in which exact order.
 2. Within the apparent vertical stripes, we can always find the discontinuities in those stripes, which means that even when some cells are very active and are more likely to fire, they still do not tend to fire after some other cells. This can be quantified as the variance along the columns in the transition matrix. We conducted a sequence order shuffle in sleep frames (preserving firing rates) and recomputed the transition matrix; we found that column variance was significantly smaller in shuffle than in the original data. This means that apart from the strong impact of firing rates, the transition matrix is also impacted by the relative order of neurons (Extended Data Fig. 9a-c).
 3. During the prediction of a sequence, we need to first predict who will fire (neuronal identity), and then predict the relative order. In that sense, we need to use the Markov model, which is rate dependent. In our original Extended Data Fig. 3D (now revised Extended Data Fig. 8), as controls, we showed that the prediction power is weak if we use forward detour preplay frames to predict theta cycles activities on T1/T3 (i.e., the not detoured tracks), or try to predict detour theta cycles with poor theta sequence. The first control showed that predicting who would fire was important, since despite the overall firing rates difference, some cells were more likely to fire in one context rather than another. The second control showed that even with the same group of cells activated on the track, their relative order would impact the predicting power.
- In other analyses where we used the Markov chain (Fig. 4g-j, Fig. 6a-f), predicting who would be active was also important in understanding the cell's participation in drifted and flickering representations. Also, in those analyses, the prediction power was compared against other sequences where the contributions from overall firing rate were similar. To make our interpretation

clearer, we added in the revised text that the result from Markov chain analysis was impacted by firing rates and the relative order.

4. In our original manuscript, we also included the tuplet motif analysis (Liu, Sibille, Dragoi, Neuron 2018; now revised Fig. 2o), which is not dependent on firing rates (tuplets were detected by comparing against shuffled sleep sequences where firing rates were preserved). As indicated by the reviewer, in the revised manuscript, we added three additional analyses that are also not firing rate dependent in terms of studying the relationship between early theta cycles and pre-run sleep:
 - 4.1. As mentioned above, we find the theta scale CCG bias correlated with CCG bias from preplay or replay for a specific track and direction, but did not correlate when we used all the pre/post-run sleep frames, due to the influence from multiple experiences onto the same sleep session, as detailed in our previous comments (Extended Data Fig. 7).
 - 4.2. For the forward detour preplay frames, we computed a pairwise order probability matrix. For example, for a pair of pyramidal units A and B, the probability of A leading B is the counts of frames in which A fires before B (used center of mass rather than the first spike, so the measure is independent from firing rates) divided by the number of frames where both cells were active. Then, in the early detour theta cycles, we got the pairwise probability of ordered pairs in each theta cycle, and we averaged the probability across all the cell pairs and theta cycles for each animal, detour session, and direction. We found the probability was significantly higher than 0.5 across samples (Extended Data Fig. 9d).
 - 4.3. We used the ensemble rank order correlation method similar to Lee & Wilson 2002 Neuron, but we applied that to compare between theta cycles and sleep frame sequences. The main limitation is that to get a significant measure, we need to have at least 5 common active cells (so we can have at least $5!=120$ independent order permutations) between a theta cycle and sleep frames. The active cell numbers in theta cycles are relatively limited compared with active cell numbers on the entire track. Across the pairs where we could run a significant test, the ratio of significant positive rank order correlation was higher than 5% chance level (Extended Data Fig. 9e). Please note that this analysis was conducted on all the pre-run sleep frames rather than selected preplay frames, since it is a frame by frame analysis that takes into account the variability of sleep frames and can identify sleep frames that were related to the experience.

Minor comments:

The use of the term “plasticity” is a bit confusing. Usually plasticity refers to synapses but that is not obvious to be the case here. Is there direct evidence that increased replay versus preplay results from synaptic plasticity, and not maybe short-term changes in membrane excitability?

This is an interesting question. In general, we follow a ‘convention’ in the field of system neuroscience where we call the experience induced circuit level change in neural dynamics as plasticity. For example, in Grosmark & Buzsáki 2016 Science, Farooq & Dragoi Science 2019, Farooq et al. Neuron 2019 plasticity was defined as the difference between replay and preplay representation of a novel experience. In Farooq et al. 2019, we showed this plasticity is not simply due to experience-driven changes in neuronal firing rates (i.e., a mix of synaptic and intrinsic plasticity) but is contributed largely by a change in cell-assembly organization of neurons post-experience, a form of ensemble temporal coding. Since our methods allow tracking plasticity at network level without distinguishing between synaptic vs. intrinsic plasticity, we used the term plasticity broadly, without a distinction between

synaptic and intrinsic forms of plasticity. In the revised manuscript, we stated the plasticity is defined and measured at circuit level rather than synaptic/intrinsic level.

There seem to be some grammatical issues where “theta sequence” (e.g. L 164), state (e.g. L350), and oscillation (L360) are used in the singular without a “the” in front.

Thank you for pointing those out. We have fixed those issues.

The need for Figure 2C-E were not clear to me. It seems these analyses are largely to confirm previous findings about phase-precession versus theta sequences that are not in dispute. If so, perhaps these should be moved to the supplement.

In the Dragoi & Buzsaki Neuron 2006 study, the dissociation between phase precession and theta sequence was shown when the linear track was familiar, the analysis was conducted by splitting the place fields into their first half and second half, and theta sequence index was computed as correlation between time-compressed temporal bias and corresponding place cell distance in space. In Feng & Foster 2015, they showed that theta phase precession and theta sequence develop in different laps. Finally, in Middleton & McHugh 2016, they manipulated CA3-CA1 synapse to see how that differentially impacted theta sequence and phase precession. Here, we showed that dissociation between phase precession and theta sequence occurs using Bayesian decoding of jittered place cell activity during novel detour runs as a main way to demonstrate that theta sequence is not simply driven by phase precession. While using a novel way to show this dissociation, this finding is indeed confirming our original 2006 finding and is consistent with the later findings from the other 2 groups. The analyses on phase locking and replay are novel. Following the reviewer’s suggestion (but please see reviewer 2 comments on this) we now moved the analysis results showing this dissociation to supplementary material and kept only the cartoon description of all 4 hypothetical drivers of theta sequence in the main figure to provide the conceptual context.

I also didn’t understand the motivation for Figures 5F-H. These could also not be put in a supplement?

We have moved them in the Extended Data Fig. 13 along with the neuronal pairwise analysis using CCG. The motivation was to show how a past experience can impact future representation at pairwise and assembly level.

The terms used to describe the different maze segments keep changing throughout the manuscript. E.g. T2L is later referred to as the Reversal Segment. Run1 is sometimes referred to as the pre-detour segment, and Run4 is sometimes called the reversal run. If this can be simplified, it would make the study easier to follow.

We acknowledge the listed challenges in defining terminology in our manuscript. We didn’t intend to change our terms throughout the manuscript. The perceived inconsistency arises because those terms have different meanings. We believe this is necessary and here are our reasons:

1. Since there are two detour sessions, terms related to temporal aspects (e.g., pre-detour and post-detour) and spatial definitions can vary depending on which detour is being referenced. As a result, we cannot use absolute terms like Run1 or T2 when describing aspects related to the general detour (i.e., both detours).
2. We have analyses such as place map correlation that requires matching the length of compared segments. During the detour session, only the middle (50 cm) segment was replaced with the detour segment (150 cm), and there was no direct correspondence between them. That required us to further refine our definition when splitting T2/T4 into two stationary 50 cm segments and the

middle segment, which can be removed and restored. T2L means the whole T2 linear track when it is not detoured, while the reversal segment is only the middle segment after the T2 detour. As stated in the original manuscript: "We called the removable middle segment during the pre-detour sessions as the 'mobile segment' of the detoured track, and when that segment was placed back during the post-detour sessions, as the 'reversal segment' relative to the detour. The other two linear tracks, T1 and T3, remained unchanged across the experiment. Throughout the manuscript, we used 'L' to refer to the 150 cm linear tracks without detour..."

We understand the relative complexity of these definitions, and we dedicated the first paragraph of the result section to clarifying them. We further refined that part, to avoid potential misunderstanding. We also added more cartoons in the main figures (Fig. 1e; Fig .4c; Extended Data Fig. 1) to increase clarity.

Labels in Fig 1D are ambiguous. What is T13? Does it mean either T1 or T3? If so why is this labelled differently in Fig 1C. What is OT in Fig 1E?

Thank you for pointing those out. We fixed those issues.

What is the running direction in Fig 1F? Or is it combined across both directions?

Thank you for pointing this out. We added the running direction in Fig. 1F example.

The arrow is not mentioned in Figure 4a legends.

We removed those arrows. They were used to help readers find animals' location on the maze.

Reviewer #1 (Remarks on code availability):

There wasn't code related to the paper and I could not install and run it.

We uploaded the related codes. The results include step by step data processing, such as preprocessing of behavior data, computing spatial tuning curves, detecting sleep frames, conducting Bayesian decoding, detecting significant preplay, computing cross correlograms, cell assemblies, and more. For the upload functions, readers can easily check and utilize them to analyze any formatted data based on the function documents.

Reviewer #2 (Remarks to the Author):

This manuscript by Zhou et al describes their study on how novel sequences within individual theta cycles (theta sequences) emerge during a novel detour experience and how they interact with sequences during awake and sleep replay (p/replay sequences) to impact place cell maps. The authors conclude that the rapid emergence of theta sequences "critically" depends on "internal dynamics" such as preplay and the flickering of both novel and familiar sequences during navigation reflects the plasticity and stability (termed "elasticity") of place maps in the hippocampal CA1. The study of p/replay in the dynamics of theta sequences and place cell maps during navigation is important. The experiments are well done and data analysis is comprehensive. However, there are several major concerns on the conclusion regarding the role of preplay, which is difficult to draw from the data currently presented.

We thank our reviewer for finding that our “study of p/replay in the dynamics of theta sequences and place cell maps during navigation is important. The experiments are well done and data analysis is comprehensive.” We have modified our statement in the manuscript and added several analyses to make the conclusion better aligned with our presented data.

1) The authors conclude that (detour) theta sequences appeared earlier than replay (Fig.2a,b). Whereas this is not surprising based on the literature, it is unclear what this comparison means, given very different definition and the different criteria for detection. Despite the authors' lengthy discussion on the equalization of false positives in their detection, it remains unconvincing because it is unknown whether same percentage of false positives should be expected from random data sets.

We agree with our reviewer that the comparison between the emergence of replay and theta sequence and the equalization of false positives are challenging tasks. Technically, it is difficult to make a comparison equal, as our reviewer pointed out. However, using a fair statistical comparison, biologically, this comparison becomes meaningful in terms of understanding how the hippocampal system develops sequential representations at different brain states. We want to discuss why we were making efforts to conduct this analysis, and what extra analyses we added in the revised manuscript to convince our readers.

1. Our goal was not to state that replay must develop later than theta sequence, as we also discussed in our response to reviewer 1. Our main goal was to investigate what could drive the rapid development of theta sequence and the compressed time scale sequence coding. We used the finding that experience-related replay statistically appeared later than theta sequence in our experiment as an argument that under normal conditions, theta sequence is not fully driven by replay, or that replay is not necessary for the emergence of theta sequence. This is consistent with the current literature, though has never been investigated directly.

2. In response to our reviewer's comment “[...] because it is unknown whether same percentage of false positives should be expected from random data sets.”, we want to emphasize that in our analysis, we obtained the false positive rate empirically from the shuffled datasets. For both replay and theta sequence detection, we ran a time bin shuffle which disrupted the temporal structures. We tested with the time bin shuffle, what was the ratio of significant replay or theta sequence under different detection criteria (Extended Data Fig. 5a-b), and defined those as false positive rates.

3. It is true that researchers in our field likely expect or believe that replay develops later than theta sequence. However, previous studies have only examined replay and theta sequences in different tasks, across different animal groups, and using unmatched detection criteria. Similarly, there are also studies claiming that disruption of medial septum can impact theta sequence but not replay (Wang, Y., Pastalkova, E., Nat Neurosci 2015, Elife 2016), while those events were not detected using the same method. We believe it is important to build a protocol where we have our best control so far over the comparison of theta sequence and waking rest replay. Our study first highlights potential concerns in drawing conclusions based on detection differences (e.g., variations in false positive rates and detection counts). However, even after accounting for these factors, we still find that during spontaneous behavior, replay develops later than theta sequences.

4. To account for potential false negative detection of replay due to under-clustered neuronal activities, in the revised manuscript we used a cluster-less decoding method (Kloosterman and Wilson, 2014), where we included ~20 times more spikes into our decoder. We detected waking rest frames based on cluster-less multi-unit activities, and re-ran the analysis for theta sequence and replay detection. We still find that theta sequence develops significantly earlier than replay (Extended Data Fig. 2c&d). Cluster-less decoding would generally reduce both false-positive and false-negative errors due to much increased sample size of data.

2) The authors conclude that theta sequences were "decorrelated" from phase precession (Fig.2d,e), because disrupting precession by jittering sometimes produced significant theta sequences. However, the raw data presented clearly show a drop in the quality of theta sequences (Fig.2d). Second, the grouped result shows that large jitters (>70ms) disrupted both precession and theta sequences (Fig2e), consistent with a tight correlation between precession and theta sequences. The authors' conclusion seems to rely on the statement that sometimes theta sequence was "significant ($p < 0.05$)" when precession was "not significant". This inference is not easy to understand. The p-values were determined differently and not necessarily linearly scaled with jittering sizes. In addition, conceptually phase precession by definition leads to theta sequences (Fig.2d top). Therefore, although there may be a contribution from non-phase-precession factors, the data presented seem more supportive for the opposite: phase precession is a major contributor to theta sequences.

We thank the reviewer for raising this important point. We were not aiming to investigate whether theta phase precession may contribute to the detection of theta sequence. We wanted to test whether phase precession alone can fully explain the early theta sequence that we observed during a novel detour. Our reviewer raised mainly three points: 1. In our analysis, jittering impacted the quality of both theta sequence and phase precession, thus implying a correlation between the phase precession and theta sequence. 2. Theta phase precession and theta sequence were measured differently and thus may not be directly comparable. 3. Conceptually, phase precession may directly lead to theta sequence (Fig. 2d top). We will address these points one by one below. Before we present our arguments, we want to point out that in the updated manuscript we changed our notation for the jittering scale from the full range to the relative change with respect to original spikes (e.g. from 70 ms to ± 35 ms) to avoid confusion. The data remained unchanged.

1. First, we agree that jittering impacted both theta phase precession and theta sequence. This was not surprising as both phenomena reflected the fine time scale features of spiking activities. Adding noise at the comparable time scale should disrupt both those features. However, showing that both theta sequence and phase precession quality correlated with the jitter level does not mean the 2 phenomena were causally related. The same would be true for a spurious relationship between phase precession and theta sequence after both processes were impacted independently by a third common factor (e.g., jittering). We now added a new analysis showing that theta quadrant ratio is negatively correlated with jittering, and the phase precession slope (negative for phase precession) is positively correlated with jittering. We ran the partial correlation between theta sequence quadrant ratio and phase precession slope, considering the confounding factor (i.e., jittering). With the confounding factor removed, theta sequence quadrant ratio was not significantly correlated with phase precession slope. Moreover, in the jittering range of ± 35 ms to ± 55 ms, where theta sequence was still significant, but phase precession was not, the quadrant ratio was not significantly correlated with precession slope (Extended Data Fig. 3). That means that in our perturbation range across animals and detour sessions, the disappearing phase precession did not necessarily cause a disappearance of theta sequence, suggesting their partial independence. The reviewer also stated "the grouped result shows that large jitters (>70ms) [now noted as ± 35 ms] disrupted both precession and theta sequences (Fig2e), consistent with a tight correlation between precession and theta sequences". We interpreted the jittering results differently, since between ± 35 ms to ± 55 ms jitter, the grouped result showed that theta sequence was significant while phase precession was not significant. We believe a potential misunderstanding might have arisen from comparing the average P value (orange line with shading) to the chance level (0.05). As we mentioned in the figure caption "Each red or blue dot represents P value from one animal, one direction, and one detour session. Binomial test results against 5% chance level are plotted with asterisks on top." For each jitter noise level there were 20 samples, and under that circumstance, a

mean P value around 0.05 suggested the number of significant samples was much higher than the chance level.

2. We agree with the reviewer's concern that P values for theta sequence and phase precession were measured differently. However, in a previous study, the statement that one phenomenon is observed while another is not, has also been made based on different measures (Feng & Foster, 2015, J Neurosci). In our case, whether the P-values scale linearly with jittering size is not a major concern. We merely created datasets where using classic detection methods and criteria in the field, researchers could conclude that theta sequence exists when phase precession does not. As we also mentioned in our response to reviewer 1, Dragoi and Buzsaki, Neuron 2006 already demonstrated this dissociation, while Feng & Foster 2015 showed that theta phase precession and theta sequence develop in different laps, and Middleton & McHugh 2016 manipulated CA3-CA1 transmission to show how that differentially impacted theta sequence and phase precession.
3. Conceptually, theta phase precession could contribute to theta sequence as shown in Fig. 2d top (now Fig. 2e, top right), but this does not necessarily imply that theta phase precession causes theta sequence. We can show that phase precession is neither a sufficient nor necessary condition for the theta sequence. For the sufficiency argument: let's imagine in our Fig. 2d top (now Fig. 2e, top right) cartoon that we shift all the spike times of cell 1 toward late (right) for a small amount, and all the spike times of cell 2 toward earlier (left) for a small amount, we can reach a state where the sequence in a theta cycle (2 leading 1) mismatch the place map sequence (1 leading 2), while if we look at single cell level, both of them exhibit theta phase precession. A similar argument and cartoon can be found in Feng & Foster, 2015, J Neurosci. For the necessity argument, our Fig. 2c top (now Fig. 2e, top left) cartoon can be a case where there are theta sequences without phase precession. We acknowledge that the biological process is not like a mathematical derivation where we can usually find strictly necessary or sufficient conditions, but it can help us understand what the contributing factors to these phenomena could be, as we observed. We emphasized in the manuscript that we are not claiming that theta phase precession makes no contribution to theta sequence. Instead, we found it can't fully explain what we observed, which is consistent with a potential contribution from preconfiguration.

3) The authors conclude that theta sequences were predicted from the preplay during sleep by a Markov process (Fig.3). However, the argument seems circular. The transition matrix (Fig.3f) was constructed from the "forward preplay" already determined. These highly selected data points were only a small proportion of activities during (pre) sleep (Fig.3c), and by definition would "predict" theta sequences, because forward preplay and theta sequences had same firing orders among cells.

We thank our reviewer for raising this important point. We understand this analysis could appear circular. We have the following arguments and new analyses to further support our conclusion:

1. The neuronal ensembles during sleep form a rich repertoire of sequence motifs where there are general backbone structures that are highly preserved across sleep sessions (Liu, Sibille, Dragoi, Hippocampus 2018) but also variable spike patterns that could support specific future representations (Farooq & Dragoi, 2024 Nature Communications). In predicting subtle features of spike activities and sequential order in individual theta cycles, we gain stronger prediction power by investigating specific frames. To show this, we also built Markov transition matrices from lap by lap place map sequences, and used them to predict activities in early laps theta cycles. We found that the prediction power can be non-significant if we use a Markov transition matrix built from behavioral time scale sequence from other tracks, thus explaining the necessity of our selection process. Please also see our response to reviewer 1 relating to why the previous CCG studies (using average activity across the whole sleep) didn't uncover preconfiguration of the theta sequence.

Predicting early detour lap theta cycle activities based on Markov model build from run laps.

a. Estimating Markov transition probability matrix (TPM) based on lap by lap place map sequence during detour run. Using the Markov model to predict activities in early detour lap theta cycles. The prediction power was compared with shuffled sequences. The plot showed the distribution of percentile vs. shuffle. The prediction power was strong as significant amount of theta cycles had larger than 95% percentile (21.93%; $P = 0$; Binomial test against 5% chance level).

b. Using the Markov model built from lap by lap run activities on other tracks to predict early detour lap theta cycle activities. Prediction power was not significant (5.51%; $P = 0.3661$; Binomial test against 5% chance level).

*** $P < 0.01$, n.s.=not significant.

2. Biologically, the detour theta cycles and detour forward preplay were related, since they represented the same context. However, technically, the analysis we employed was not really circular. The detour preplay frames were selected using Bayesian decoding that computes posterior probability for multiple individual 20 ms time bins with no assumption of their sequentiality, built using an average (spatial) place map computed across the whole detour session at behavioral time scale. Thus, there are several differences between the computation of preplay and those of theta sequence. First, the activities in the early theta cycle didn't perfectly match the average place maps in the first 2 laps; only ~60% of theta cycles depicted positive quadrant ratios. Second, the decoding process detecting preplay was conducted between space (run place maps) and time (sleep frames), while the Markov model compared the sequences between two temporal structures (theta cycles and sleep frames), without using spatial information. Third, the decoding process compares behavior time scale place map with compressed time scale activities in sleep frames, while the Markov model directly investigated sleep frame and theta cycle activities at compressed time scales. Fourth, the preplay frames were selected using Bayesian decoding, which considers all the spikes, while in predicting sequences in the Markov model, we used the sequence computed based on the center of mass for each spike, which is more similar to a rank order approach to study sequential structure. Importantly, in our data, when detecting frames significant as preplay or replay for detour experience using Bayesian decoding on the one hand and rank order Spearman correlation on the other hand, we found the 2 groups of frames were uncorrelated, with only 0.34% (preplay) or 0.39% (replay) of frames being significant for both detection methods (Extended Data Fig. 9f-g; see also Tingley & Peyrache 2022). Thus, the approaches we used to show prediction of the theta sequence from preplay are not circular; they, however, enabled us to identify specific frames within a repertoire of sequential motifs during sleep.

3. In the original manuscript, we used a tuple analysis (Liu, Sibille, Dragoi, Neuron 2018; now revised Fig. 2o) where the tuples were detected from all sleep frames. Now, we also added a template matching type of analysis, which does not require extra selection of sleep frames. We directly conducted a rank order correlation between spike sequences in early lap theta cycles and pre-run sleep frames when there were at least 5 common active cells (so we can have measures for significance). Across those pairs, we found the ratio of significant positive rank order correlation was higher than 5% chance level (Extended Data Fig. 9e). This supports our conclusion of compatibility between network pre-configuration and activity during early theta cycles during the novel detour.

4) *The conclusion in the abstract saying the internal dynamics "critically support" rapid sequence coding. The phrase is too strong for the data presented. It is unknown whether preplay was causal or even contributed equally as novel experience.*

We agree with the reviewer that both preconfiguration and experience contribute to the compressed scale sequential representation. We mentioned in the original manuscript that "This suggests that pre-existing short sequence motifs contributed a backbone for the rapid expression of time-compressed sequential coding during novel detour exploration in conjunction with plasticity-driving inputs from presumed specific sensory-motor external stimuli."

We also added new analysis showing how Markov prediction power increased from pre-run sleep to post-run sleep as evidence of experience induced plasticity in post-run sleep compared to pre-run. We also found that pre-run sleep had stable prediction power over theta cycles in early, middle, and late laps, while post-run sleep had increasing prediction power from early to late experience (Fig. 2n). This analysis highlighted the different temporal dynamics of how preconfiguration and experience shape the compressed scale sequence coding.

We adjusted our abstract to make the statement better aligned with our findings and removed the word 'critically'.

5) *In many of the plots, it is unclear what each dot (sample) is.*

We thank our reviewer for pointing that out. Now in all the statistical plots in main and extended figures, we have specified in the figure caption what each dot (sample) represents.

6) *Fig.3e, were there 57 cells? But the line thickness is not consistent with such a big number.*

There are 57 cells for that example animal; only part of those were active in these example pre-detour sleep frames. We modified the y-axis ticks to avoid confusion, and we also adjusted the marker size to increase clarity.

Response to reviewers' comments for manuscript NCOMMS-25-06854-T

Reviewer #1 (Remarks to the Author):

The authors provided a revision that rebuttal that nicely answered some of my concerns. I thought Fig 2h was a nice addition, and the removal of some of the panels makes the main figures more straightforward. And I appreciate they clarified the segments and their overlaps in Fig 5 analyses. The controls they did for the Bayesian analysis in Fig 6 appear quite reasonable. They also provided a good explanation that the Markov method does capture something other than just firing rates since it only works on some parts of the maze. However, for me some critical issues remain.

We are glad that the reviewer found most of the previous concerns satisfactorily addressed and we greatly appreciate the reviewer's effort in helping us improve our manuscript. In response to the new concerns raised, we have conducted additional analyses and provided several clarifications. We believe these modifications will not only address our reviewer's comments but will also enhance the clarity and rigor of the manuscript for the broader audience.

*First, with regards to why the CCG doesn't work in this task, their response makes sense: the rat runs in two direction which cancel out any directionality in the firing order and there are many maze sections that have different patterns. "method that averages across the whole sleep reflecting multiple experiences, like CCG, fails to capture the one experience-related sequential structures that are only expressed in a part of sleep frames (6-7%)". "However, I had not realized that so many of their arguments regarding "prediction" relied on this subselection of "forward detour preplays". This really supports the second reviewer's major concern about the circularity of the related claims. If 6 percent of events are like the theta sequences but another 6 percent are the exact reverse running direction, then I don't think anything can be said to have been predicted from these periods combined. I read carefully the authors rebuttal on this point, but unfortunately I did not find those arguments persuasive. E.g. I don't understand the authors assertion that preselection of forward preplays makes "no assumption of their sequentiality". If the overall activity does not reflect any particular temporal pattern, then to me it doesn't make sense to suggest there are existing network biases that are predictive of the later theta sequences. I think it could be fine to detect what proportion of events in sleep, etc, qualify as preplay and report the proportions, as the authors do in many points, and to report these patterns as being *present* before the detour segment. However, the claim that these patterns were *predictive* is indeed circular, since they could not be detected without using the specific place-field sequence on the detour.*

We agree with our reviewer's comment that, since our analysis relies on place cell sequence information from the future detour run, it does not constitute an absolute prediction of specific theta sequences from the overall preceding sleep patterns. However, we believe our approach goes beyond simply assessing the presence of theta sequence structure in the pre-detour sleep. First, we did not directly use information from theta sequence to test if it is present in the preceding sleep, but instead use the place map on the detour to select detour representation frames from an otherwise rich sleep repertoire. As the reviewer noted, such selection is necessary given the variability of run sequences in the detour and the large repertoire of activity in sleep frames/events, which can mask structure when averaging across all sleep events. Our model is that a repertoire of different theta sequences can be tracked and matched to a repertoire of preplay sequences in the preceding sleep. If the correspondence is missed, the theta sequence-preplay relationship will be undetected, while still present. The theta sequence is correlated with the corresponding place cell sequence only to a certain and variable degree (e.g., they are weakly correlated in the first lap, less than 20% of theta sequences are significantly matching place cell sequences overall), but it is not the theta sequence itself. An undeniable circularity would be to directly use theta sequences during detour to look for their correlates among theta sequence-correlated frames of activity during sleep, but that is not something we did in

our study. Second, we ran our Markov chain predictive algorithm and tuple analysis to predict theta sequences from corresponding sleep frames. The *corresponding* aspect of frames does not guarantee the success of the prediction algorithm (i.e., Markov chain and tuple analyses). In this sense only, we claimed that these patterns of activity in the pre-detour sleep can be *predictive* of future theta sequences, without implying that the entire sleep session is indiscriminately predictive of a particular theta sequence. One can also envision a scenario in which these 2 analyses do not reach significance for prediction, in which case we would not call them predictive. Third, the activity during sleep frames detected using Bayesian decoding and rank order correlation are not well correlated and defy circularity since Bayesian decoding uses all the spikes while rank order correlation or the Markov model use single center-of-mass spike time per neuron. Fourth, multiple different theta sequences can be *predicted* as defined above from multiple different sets of sleep frames, making this phenomenon a general one, suggesting that, overall, pre-detour sleep patterns can be predictive of future theta sequences on detour. The frame selection process also allowed us to reveal both the grouping of neurons into assemblies and their temporal relationships in representing a novel trajectory. We acknowledge our reviewer's concern regarding the use of the terms *predictive* or *prediction*, thus we clarified our use of prediction in the revised manuscript, Lines 294-300: "Importantly, our analysis didn't prove an absolute prediction of novel theta sequence, since we used the future information of detour place map to select the corresponding sleep frames. This selection procedure was necessary considering the large repertoire of sequential activity patterns expressed during sleep and the variability of theta sequences during run (Extended Data Fig. 8a-f). Thus, the "prediction power" in our analysis revealed the compatibility between the corresponding preconfigured frames of activity during sleep and the future time-compressed theta sequence, rather than an absolute prediction of theta sequence from indiscriminate sleep patterns.". We thank the reviewer for highlighting this point.

In the rebuttal of this point, the authors also refer to some tuple analysis from an earlier paper. I read this paper and really tried but it was very difficult to understand. I still don't really understand what is a 2-neuron tuple and how it differs from the pairwise CCG peak. If two cells reliably fire in some order, that should be very clear in the CCG. What does it the tuple mean then? Maybe it's picking up misclustered bursts or something other than reliable patterns?

Our reviewer is correct that both 2-neuron tuple and CCG analysis study the pair-wise relationship of neuronal pairs. Below, there are several differences that we are going to clarify along with some new analyses to support our original arguments:

1. Conceptual distinction between CCG and tuple analysis:

Both methods probe pairwise neuronal interactions, while the key distinction lies in what aspects they emphasize: CCG focuses on precise spike timing relationships across time lags, capturing temporal asymmetries (e.g., neuron A tends to fire before B). On the other hand, the 2-neuron tuple analysis considers not only the temporal relation but also this pair's tendency to be active within the same frame, related to understanding of who tends to be recruited together and in what order during the future run. The 2-neuron tuple analysis can reflect co-active features in the sense that both A->B and B->A can be detected as different tuples. From now on, we will call them bidirectional tuples and call the others unidirectional tuples. This is technically possible because we detected tuples by comparing the occurrence of sequential structures in individual frames against shuffle sleep where we only preserve the firing rates. The occurrence of A->B and B->A can both exceed chance level if neuron A and B are more likely to be active in the same frame. We have included this clarification in Lines 329-334 as well

as in the Methods. We quantified the distribution of 2-neuron tuples and find in our datasets that 35.78% are bidirectional (17.89% as a mirror copy of another 17.89%; Extended Data Fig. 11a).

2. Relation between tuples and CCG temporal bias:

To further study how the tuplelet analysis is related to CCG analysis, we counted the occurrences of all the 2-neuron sequential structures in forward and reverse order, and we defined the forward ratio as counts of forward sequence divided by counts of co-active sequence (forward+reverse). Bidirectional tuplelets would have a forward ratio closer to 0.5 while unidirectional tuplelets would have a forward ratio closer to 1. We found the forward ratio was positively correlated with the temporal bias computed from CCG. However, the tuplelet analysis can detect bidirectional tuplelets, which have a CCG temporal bias close to 0. The latter aspect would blur the CCG analysis findings of temporal relationship between neuronal pairs.

3. Are bidirectional tuplelets meaningful?

We tested the functional relevance of bidirectional tuplelets. We utilized the fact that during the detour run, due to the relatively high bidirectionality of the detour place map, many cell pairs exhibited reversed order across two run directions. We found those bidirectional cell pairs during run are significantly enriched among the bidirectional pre-detour run sleep tuplelets (Extended Data Fig. 11b-c). Thus, the flexibility in relative order of cell pairs during sleep may facilitate their flexible sequential order during run, while the fact that they tend to cofire within the same frame could support their co-recruitment during the future representation.

4. Methodological note on temporal precision:

Another difference is that CCG analysis uses all spikes, while tuplelet analysis collapses spikes within an event frame into a single center-of-mass spike time per neuron. This reduces temporal precision but aligns with ensemble-level co-activation, which may explain some differences between the two methods (CCG and tuplelet).

5. On the misclustering concerns:

Earlier during the clustering process, we checked the CCG between all the cell pairs within the same tetrode; the misclustered bursts should exhibit a CCG pattern with sharp peaks at around +5 or -5 ms with clear refractory period around 0 ms. We merged those units if we observed this pattern; thus, we are not concerned about the potential misclustering of bursts in our analysis.

6. On predictive power and limitations:

Roughly 25% of 2-neuron sequences during run are accounted for by pre-run sleep tuplelets. While this is statistically significant, it leaves ample room for learning and sensory-driven dynamics to shape run sequences. The modest effect size, coupled with the CCG's lower sensitivity to bidirectional patterns, may also be the reason why CCG analysis didn't exhibit strong pre-configuration.

The rebuttal also provide some new rank-order correlations between pre-sleep and theta. This analysis is promising but seems preliminary and should be better illustrated. It seems concerning that the rank-order correlation does not do very well in identifying the Bayesian decoded preplays and replays, since Bayesian decoding is at the heart of this paper. It also left unclear how both the forward and reverse detour directions could be matched to these preplays.

Related to our reviewer's first concern "that the rank-order correlation does not do very well in identifying the Bayesian decoded preplays and replays", we understand it as a concern over a general issue related to p/replay study rather than a specific issue related to our data or analysis. Our explanation to this is that the 2 methods (Bayesian decoding and rank-order correlation) are measuring different aspects of p/replay; while neuronal ensemble activities are high dimensional, we usually use one value/measure to answer the question "how do these sleep neuronal activities match the run representation?" Rank-order correlation uses one entry/neuron in space (e.g., location of place field peak) and one in time (e.g., center-of-mass spike time during a sleep frame) and emphasizes neuronal sequentiality, while decoding uses all spikes in the frame and all place map activity on the track and decodes the representation relying on neural coactivation independent of neuronal sequentiality. These methods are inherently information compression processes which emphasize some features but inevitably ignore other features. For example, even when we ask the simple question "how similar are two vectors?", we can have different measures such as Pearson's correlation, rank order correlation, cosine distance, Euclidean distance, while the results from Pearson's correlation will be independent from the results from Euclidean distance. However, if the general phenomenon is strong enough (e.g., replay, preplay), we should be able to observe it with several methods emphasizing different aspects of the phenomenon. In that sense, we gain more confidence in our results by supporting the conclusion with different methodologies.

On our reviewer's second question "how both the forward and reverse detour directions could be matched to these preplays", the reviewer is likely asking how detected preplay can match place maps with two opposite run directions. First of all, similar to the Bayesian decoding, the rank-order correlation is also a frame-by-frame analysis, meaning we can detect which frames are preplays for each run direction. Preplay frames for each run direction can distribute among all the frames differently and may not overlap across two run directions. We have included this clarification in Lines 321-323. Second, if the place map is strictly bidirectional (which is not the case in real data), then the forward p/replay (correlation higher than 95% of shuffle) for one direction would just be the reverse p/replay (correlation lower than 95% of shuffle) for another direction. Both forward and reverse trajectories were considered as p/replay events in many studies where they were assumed to carry distinct functions. In this analysis, we only focus on forward p/replay exhibiting the match of sequence between theta cycles and pre-detour sleep frames given theta sequence is mostly forward.

Another point I thought Reviewer 2 had an excellent comment about the the relationship between phase precession and theta sequences. To me it sounds like the jitter mainly tells you about the temporal sensitivity of the measures used. So if phase-precession measures used are very sensitive to jitter (maybe because they pool all trials instead of using single trials—that has been found that single trials are more robust than averages) and also perhaps because of sawtooth shape of theta waves impacting phase measurements with the Hilbert transform (Belluscio et al), then it could be that small amounts of jitter ruins phase precession but not theta sequences. But that doesn't mean that the two phenomena are unrelated (even if that kind of argument was used in some papers before). One is likely just based on measures that more sensitive to jitter than the other.

We acknowledge the difficulty in comparing phenomena with different detection methods and we agree with our reviewer's comment that detection methods can have different sensitivities to noise. We want to emphasize that our main point is not to state that theta phase precession is not related to theta sequence. We believe phase precession relates to theta sequence, but does not fully account for the development of theta sequence. In support of this idea, we created a scenario where there are theta sequences present but not phase precession.

To further address reviewer's concern and to emphasize the fact that theta sequence is an ensemble phenomenon while theta phase precession is a single unit property, we conducted a new analysis. During the first two laps on the detour segment, for each pyramidal cell, we only kept one spike or spikes within bursts (we keep as many spikes as possible as long as the maximum ISI < 20 ms) per lap, per place field. Under that condition, we eliminated the single cell phase precession in the first 2 laps and ended up with a positive correlation between spike theta phase and position in the field due to bursts. After this process, we still found significant theta sequences based on decoding in the first 2 laps where the decoded trajectory swept from past to the future (Extended Data Fig. 4). This result supports the idea that theta sequences can emerge at the population level even in the absence of detectable single-cell phase precession, likely due to precise temporal coordination among assemblies. A more compelling argument is that phase precession plays no role in selection of which several distinct neurons will be activated within a theta cycle together as theta sequence while preplay/tuplet analyses indicate this selection/grouping was already present within and could be predicted from selected frames of preplay activity.

Around L379, the study concludes "the novel detour differentially impacted the associated T2/T4 tracks stronger compared to the non-detoured T1/T3 tracks suggesting its experience was different than a simple exposure to a novel track environment." However, the study reports that there were actually more elastic cells, defined as those with place-fields that return, on T2/T4 than on T1/T3. (Note also that the claim on L382 about the firing rates is only true for T2/T4, not T1/T3). This suggests a rather surprising lasting effect of the detour on the other tracks. So either the plastic/elastic measures are problematic or this conclusion is over simplified.

We thank our reviewer for pointing out this potential confusion. We now clarify that the plastic and elastic cells only belong to a subpopulation if those cells experience major remapping from pre-detour session to detour session. T1/T3 had fewer elastic cells just because they experienced weaker remapping during the detour session. Based on our formula to compute plastic/elastic measure, plastic/elastic measure = detour vs. post-detour similarity minus post-detour vs. pre-detour similarity, we can imagine that a stable cell throughout sessions will have a measure close to 0, which is not plastic (plastic/elastic measure = 1) nor elastic (plastic/elastic measure = -1). To directly measure the lasting impact of detour on T2/T4 or T1/T3, we can use the place map similarity between post-detour session vs. pre-detour session, where T1/T3 are more stable (Fig. 4e). The concept of plastic/elastic cell is derived from a different perspective: given cells have strong remapping (as shown in Fig. 1d-h), what would be their representations during the reversal run. In the revised manuscript, we further clarified this to avoid potential misunderstanding (Lines 399-403).

One other point of confusion for me. The study notes 5 rats with 2 detour sessions per rat, summing to 10 detour sessions. And yet there are often 16 or more data points in the figures. What am I missing? Maybe it would help if the figure legends made note of the n's used in the various panels.

In our manuscript figure captions, we specify what each dot represents. In most of cases, "each dot representing one animal, direction, and detour session" thus we will have 5 (rats) * 2 (directions) * 2 (detour sessions) = 20 data points. We treat 2 directions as separate points as place maps are not strictly bi-directional. This is a common practice especially in studies related to forward and reverse p/replays. Otherwise, if the place map is strictly bidirectional, forward p/replay in one run direction would be reversed p/replay in another direction. In analyses related to sleep, we excluded Rat4 due to

insufficient sleep frames and had 16 data points. Now we have added “n” to all the figure captions where there were sample data points and statistical tests.

We have also updated Extended Data Fig. 16 figure caption. Now it is “with each dot representing one animal, one direction and one detour session”. In the original manuscript we averaged results across directions in flickering analysis (Fig. 5c&i) and now we treat directions as separate data points to match the style in the other analyses. The conclusion remains the same.

Finally, the manuscript makes a distinction between its findings and those reported by Feng, Silva, and Foster J Neurosci 2015. I looked back at this paper, and I noticed that they emphasized the rapid improvement in the theta sequences after the first lap, but did not report on whether or not the first lap itself was also significantly ordered. Indeed the first lap quadrants shown in that paper are not much different from those in this one. This could be an important detail in reconciling the two studies.

We appreciate this valuable insight from our reviewer. We checked that paper and found that despite the claim that “distinct theta sequences were absent on the first lap”, they only conducted analysis to compare lap1 vs. other laps rather than test the significance of lap1, as our reviewer pointed out. We also want to point out one important technical detail: in their analysis, the authors probably only corrected decoding positions once based on the animal's position at the center of theta cycle rather than correcting independently for each time bin. In their Fig. 1b and Fig. 2a we can clearly observe the decoding trajectory started below position 0 in the first theta cycle while it started slightly above position 0 in the last theta cycle. That could lead to a problem in computing the quadrant ratio even only using the middle theta cycle, as the actual trajectory of the animal traverses from past to the future. That means that even if the decoding probability strictly matches true position without having any theta sequence structure, they will detect a positive quadrant ratio just due to the movement of the animal. That can be the reason why they didn't report the quadrant ratio quantification for each lap. In our dataset, even after the independent time bin correction, we still observed the significant positive quadrant ratio in lap1. Nevertheless, when citing their study, we prefer to refer to their conclusions as stated in their Abstract.

Reviewer #1 (Remarks on code availability):

I just glanced through the code quickly. Seems to be appropriately detailed.

We thank our reviewer for checking the codes.

Reviewer #2 (Remarks to the Author):

This revised manuscript has mostly addressed my questions. Although I feel like some of the responses are not super clear, I think the revision is adequate.

We thank our reviewer for their support of our work during the review process.